# Amorphous alloys surpass E/10 strength limit at extreme strain rates

Wenqing Zhu [1,7], Zhi Li [2,7], Hua Shu[3], Huajian Gao [2,4,5,8] ✉ &
Xiaoding Wei [1,6,8] ✉

Theoretical predictions of the ideal strength of materials range from $E/30$ to $E/10$ ($E$ is Young's modulus). However, despite intense interest over the last decade, the value of the ideal strength achievable through experiments for metals remains a mystery. This study showcases the remarkable spall strength of $Cu_{50}Zr_{50}$ amorphous alloy that exceeds the $E/10$ limit at strain rates greater than $10^7$ s$^{-1}$ through laser-induced shock experiments. The material exhibits a spall strength of 11.5 GPa, approximately $E/6$ or 1/13 of its P-wave modulus, which sets a record for the elastic limit of metals. Electron microscopy and large-scale molecular dynamics simulations reveal that the primary failure mechanism at extreme strain rates is void nucleation and growth, rather than shear-banding. The rate dependence of material strength is explained by a void kinetic model controlled by surface energy. These findings help advance our understanding on the mechanical behavior of amorphous alloys under extreme strain rates.

The pursuit of materials with ideal strength is a long-term goal of scientists working in the fields of materials science and mechanics who are interested in intrinsic properties. After the ground-breaking work by Frenkel[1], the ideal strength of materials was estimated by the well-known $\frac{E}{N}$ rule, in which $E$ is Young's modulus; $N$ is a constant with a value of approximately 10 (ref. 2). With the advances in nanofabrication and nanomechanical testing, strengths near the theoretical limit have been reported from experiments on nanosized crystalline quasi-brittle materials, including silicon nanowires ($E/7$)[3], carbon nanotubes ($E/10$)[4], graphene ($E/9$)[5], and diamond nanoneedles (~$E/10$)[6,7]. However, whether the measured strength of metallic materials can reach a comparable level remains unclear.

Unlike quasi-brittle materials, single-crystal and multi-grain metals deform plastically via dislocation mechanisms that are strongly affected by vacancies, impurities, twins, and grain boundaries[2]. Thus, reducing the sample size alone does not necessarily

ensure that metals achieve their ideal strengths. Richter et al. reported a strength value of ~$E/25$ for single-crystal Cu nanowhiskers[8]. Chen et al. reported a strength of ~$E/18$ for single-crystal Pd nanowhiskers[9]. Kim et al. revealed the failure mechanism of thin-layer twin formations for <110> Al nanowires with a strength of ~$E/23$[10]. Researchers have also attempted to push the strength limit using ultrahigh strain rate tests where specimens deformed under the uniaxial strain condition[11–13]. Under this condition, the spall strength is compared with the pressure wave (P-wave) modulus $M = \rho c^2$ in which $\rho$ and $c$ are the material density and the sound of speed, respectively. Jarmakani et al. conducted laser-driven shock tests at strain rates of $\dot{\varepsilon} \sim 2 \times 10^6$ s$^{-1}$ and reported a spall strength ~$M/25$ for vanadium monocrystals[14]. de Rességuier et al. reported a spall strength ~$M/30$ for single-crystal Mg at $\dot{\varepsilon}\sim1 \times 10^7$ s$^{-1}$ (ref. 15). Coakley et al. conducted laser ablation on polycrystalline Cu foils at strain rates of $\dot{\varepsilon}\sim5 \times 10^8$ s$^{-1}$ and reported a spall strength of ~$M/25$[16]. Righi et al. reported high spall strengths for

[1]State Key Laboratory for Turbulence and Complex System, Department of Mechanics and Engineering Science, College of Engineering, Peking University, Beijing 100871, China. [2]Institute of High Performance Computing, Agency for Science, Technology and Research (A*STAR), Singapore 138632, Republic of Singapore. [3]Shanghai Institute of Laser Plasma, China Academy of Engineering Physics, Shanghai 201800, China. [4]School of Mechanical and Aerospace Engineering, College of Engineering, Nanyang Technological University, 70 Nanyang Drive, 637457 Singapore, Republic of Singapore. [5]Center for Advanced Mechanics and Materials, Applied Mechanics Laboratory, Department of Engineering Mechanics, Tsinghua University, Beijing 100084, China. [6]Peking University Nanchang Innovation Institute, Nanchang 330000, China. [7]These authors contributed equally: Wenqing Zhu, Zhi Li. [8]These authors jointly supervised this work: Huajian Gao, Xiaoding Wei. ✉e-mail: gao.huajian@tsinghua.edu.cn; xdwei@pku.edu.cn

single-crystal (~$M$/26) and nanocrystals of Fe (~$M$/42) at strain rates of ~$2 \times 10^7$ s$^{-1}$ (ref. 17). Even though the strain rates were many orders of magnitude higher than in the quasi-static experiments, the material's normalized strength (by Young's modulus or pressure wave modulus) is still notably less than 1/25. This is because single-crystal or nanocrystalline metals undergo extensive plastic deformation mediated by dislocation movements, grain rotation, and grain boundary sliding (predominant at low or moderate strain rates[18,19]) and mechanical twinning (predominant at ultrahigh strain rates[17]) prior to failure, which accounts for the notable difference between experimental and ideal strength values.

Metallic glasses (MGs) are renowned for their exceptionally high elastic limits and strengths due to their amorphous nature, which prevents the classic plastic deformation mechanisms[20,21]. Even under quasi-static loading, the mechanical strength of bulk MGs can reach $E$/50[22]. Subsequently, Tian et al. achieved strengths of ~$E$/20 for Cu-Zr nanowires[23]. This is because in nanosized MGs, it is difficult to launch shear banding, the main failure mechanism of MGs at low strain rates, due to the reduced number of clusters of shear transformation zones (STZs)[23,24]. Tang et al. conducted plate impact tests on MGs to achieve higher strain rates (~$10^6$ s$^{-1}$)[25]. The spalling at fracture surfaces of MGs showed cup-cone structures, indicating a mixed failure mechanism of cavitation and local shear banding. Nevertheless, Tang et al. obtained the spall strength of approximately $M$/38 (or ~$E$/21). Thus, $E$/20-$E$/25 has long been regarded as the upper limit of the measurable strength for metals. In contrast, the strength and the mechanisms of failure for MGs at extreme strain rates remain largely unexplored.

In this study, we test Cu$_{50}$Zr$_{50}$ at strain rates $\dot{\varepsilon}>1 \times 10^7$ s$^{-1}$ to assess its mechanical properties under these extreme conditions. The spall strength of the material reaches 11.5 GPa, approaching approximately $E$/6. The experimental observations, complemented with large-scale molecular dynamics (MD) simulations and continuum models, reveal that the amorphous material fails primarily due to void nucleation and growth, rather than shear-banding.

## Results and discussion

Thin Cu$_{50}$Zr$_{50}$ MG discs (50–100 μm thick) were fabricated using the single-roller melt-spinning method. We hit the front surface of each sample using a nanosecond Nd:glass laser (Shanghai Shenguang-II laser facility, National Laboratory on High Power Laser and Physics in Shanghai, China) to generate a shock (Fig. 1a). Two laser pulse durations (1 and 2.5 ns) and laser energy inputs ranging from 2 to 15 J were employed. A line image velocity interferometer system for any reflector (VISAR) was used to measure the rear free surface velocity (FSV) and deduce the spall strength and strain rate; a detailed analysis

is given in Materials and Methods. Supplementary Fig. 3a and b illustrate the representative FSV curves for the two pulse durations. The peak values of FSV ($v_{fsp}$), 870–1750 m/s, were significantly greater than those previously reported using the plate impact approach (300–600 m/s)[25–28]. The tensile strain rates reached $1.4-2.8 \times 10^7$ s$^{-1}$ (see Supplementary Fig. 5). At high energy inputs, spallation occurred as the result of the interactions between incident and reflected shock waves, and the corresponding spall strengths varied from 6.6 to 11.5 GPa (Fig. 1b). The highest spall strength was approximately $M$/13 (the measured P-wave modulus $M$ = 149.5 GPa, see Supplementary Table 1). Note that the Young's modulus $E$ = 65.9 GPa(measured by uniaxial tensile test; the corresponding Poisson's ratio $v$ = 0.407, agreeing with the literature[29]), thus, the material strength approached approximately $E$/6. To the best of our knowledge, this was the first time that the measured strength of metals surpasses $E$/10 and approaches $M$/10. Notably, this record-breaking strength was achieved in the absence of the strengthening mechanisms present in crystalline metals, such as dislocation interactions and grain boundary strengthening. It is crucial to identify the underlying mechanism that grants the amorphous alloys exceptional mechanical strength.

First, we performed fractography on the spall plane. Scanning electron microscopy (SEM) images showed dimples rather than the cup-and-cone features on the spall plane (Fig. 2a, b and Supplementary Fig. 7a). More interestingly, the dimple size showed strong strain rate dependence (Fig. 2 and Supplementary Fig. 7). At $\dot{\varepsilon}= 1.9 \times 10^7$ s$^{-1}$ (5.0 J laser power and 2.5 ns pulse duration), the dimple diameter ranges from 2 to 3 μm. In contrast, at $\dot{\varepsilon}= 2.8 \times 10^7$ s$^{-1}$ (14.3 J laser energy and 2.5 ns pulse duration), a hierarchical dimple structure was seen; the majority of dimples had diameters of several hundred nanometers and only a small percentage exceeded one micron. Using a focused ion beam (FIB), we cut into the spall surfaces to examine the void distribution in the thickness direction. The specimen shocked by the 5.0 J laser pulse contained micro-voids scattered within a few microns beneath the spall surface (Fig. 3). In contrast, the specimen shocked by the 8.4 J laser pulse contained interconnected nanovoids underneath the spall surface. To further confirm cavitation is the main failure mechanism in our tests, we milled into an un-spalled specimen tested by a less intense laser pulse (1.3J energy and 1 ns duration) using FIB. At a depth of approximately 5 microns from the back surface, we also discovered features of cavity initiation and coalescence (Supplementary Fig. 8). Last, X-ray diffraction (XRD) and selected-area electron diffraction (SAED) in a transmission electron microscope (TEM) were carried out to assure that the amorphous nature was maintained in the specimens after tests. XRD spectra of the two post-test specimens revealed the same broadened diffuse humps as in the as-cast specimens. SAED on

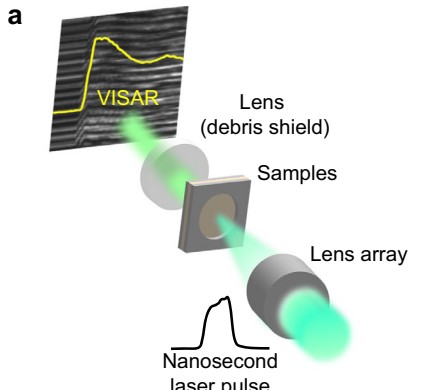
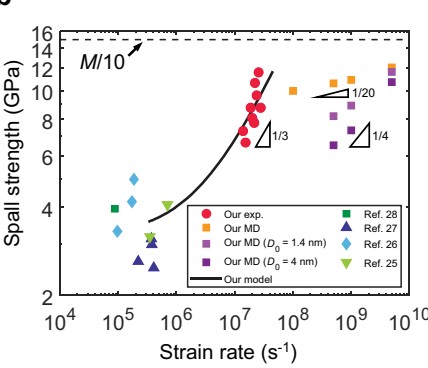

**Fig. 1 | Laser-induced shock tests of Cu-Zr MG disks. a** Schematic diagram of the laser-induced shock experimental equipment. **b** Summary of data for spall strength vs. tensile strain rate (red circles). The results for Zr-based MGs from previous reports are included for comparison (refs. 25–28). Yellow and purple squares are

the results obtained from large-scale MD simulations with or without initial nanovoids, respectively. The solid line is the prediction of the rate-dependent strength based on our kinetic model of void growth (Eqs. (1) and (2)). The dashed line indicates the limit of M/10.

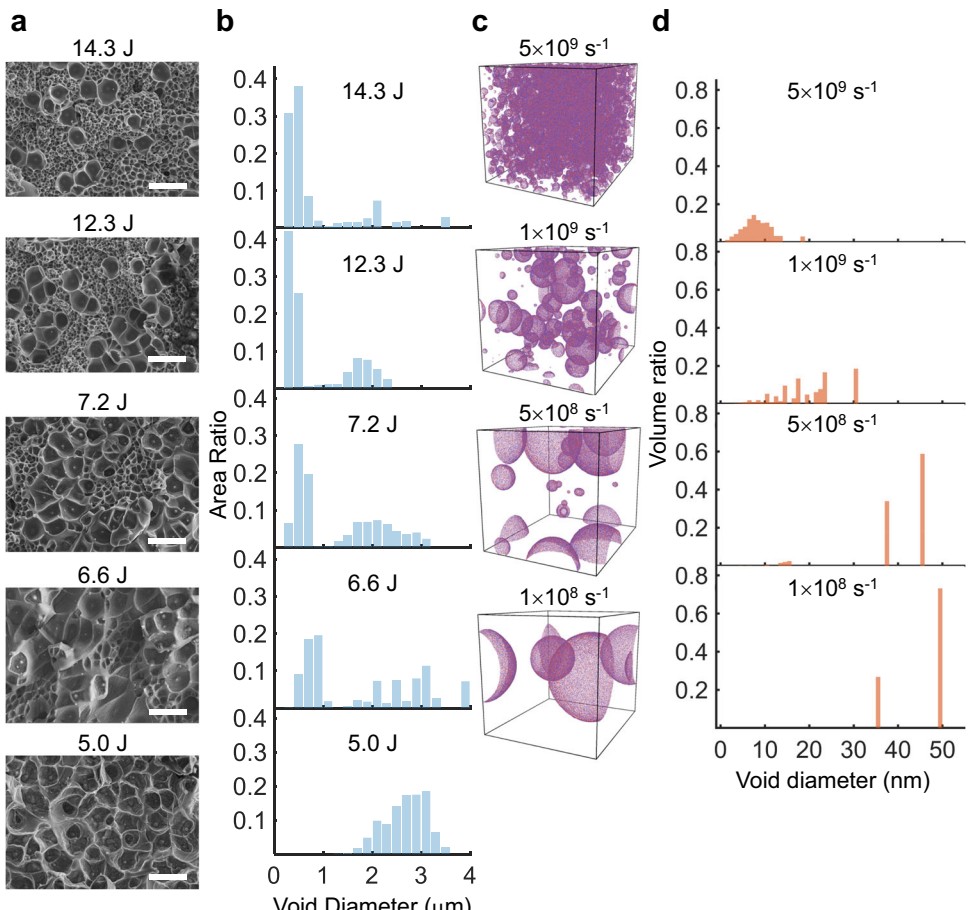

**Fig. 2 | Rate-dependent void size distribution on the spall plane. a** SEM micrographs of the spall surfaces of the samples tested by the laser with 2.5 ns pulse duration and different input energies (from bottom to top: 5.0 J, 6.6 J, 7.2 J, 12.3 J and 14.3 J). Scale bar: 5 μm. **b** Histograms of the void size distributions at different laser powers, i.e., strain rates. **c** Final void morphology obtained from large-scale MD simulations for $Cu_{50}Zr_{50}$ stretched at various strain rates. **d** The corresponding statistics for the void sizes obtained in MD simulations.

different spots below the spall plane also only showed diffuse halos, consistent with the disordered atomic structures observed in high-resolution TEM (HRTEM) images (Supplementary Fig. 9).

The above characterizations indicated that no crystallization or related crystal plasticity mechanism happened during or after the tests, and MG failed predominantly due to void growth and coalescence rather than shear banding. The main factor differing our tests from previous ones is the ultrahigh strain rate. It is well-known that intrinsic spatial heterogeneity is essential to the unique structure–property relationship of MGs[30–34]. MGs consist of stable regions where atoms are more densely packed and rheological regions where atoms are more loosely packed, i.e., defective spots. At low and moderate strain rates, these defective spots distort and serve as nucleation sites for shear banding, i.e., STZs[35]. However, the laser pulse durations in our study, 1 and 2.5 ns, are substantially shorter than the timescale for shear band initiation, which is typically from several to tens of microseconds[36,37]. Thus, at extremely fast strain rates, these defective spots serve as cavitation nucleation sites, also known as the tension transformation zones or TTZs[38]. Moreover, the higher the strain rate is, the greater the number of TTZs that are activated. This is because increasing the tensile hydrostatic stress decreases the free energy barrier for cavitation in MG, as suggested by Guan et al.[39]. We note that the spall strength of materials may be influenced by the compressive stress amplitude, which is coupled with the strain rate. Nonetheless, a recent study suggests that the spall strength of a Zr-based MG is barely dependent on the stress amplitude[40]. Thus, it is reasonable to argue that the strain rate is the primary determinant influencing the material strength.

Large-scale MD simulations of the uniaxial strain tension of $Cu_{50}Zr_{50}$ MG at strain rates in the range of $1 \times 10^8$–$5 \times 10^9$ s$^{-1}$ offer more insight into microscopic material failure mechanisms. First, the ultimate strength of the Cu-Zr MG increased with the strain rate, which was consistent with the results of our laser shock tests (Fig. 1b and Supplementary Fig. 10c). The strain rate sensitivity of the material strength obtained from MD simulations was notably less than that found in our experiments. This could be attributed to the fact that the atomistic model was still substantially smaller than the real specimens. In addition, the extreme cooling rate used for generating the atomistic model ($1.7 \times 10^{12}$ K s$^{-1}$) was significantly faster than those in melt spinning ($10^4$–$10^6$ K s$^{-1}$), which could result in different nonequilibrium states between the atomistic model and real specimens[41]. For instance, the real specimens might contain scattered "defects" such as nanopores[42], while the MD model is rather "ideal". Thus, we also generated two "defective" atomistic models that contained nanovoids with diameters of 1.4 and 4 nm, respectively, by removing the atoms in the center. Applying uniaxial strain tensile tests on these two models revealed that the strain rate sensitivity was highly sensitive to the flaw size—the model with a smaller nanopore showed a strain rate sensitivity of 0.16, while the one with a larger nanopore showed a strain rate sensitivity of 0.22, both of which were significantly greater than the ideal model (Fig. 1b). We carried out additional computational studies of the effects of cooling rate and potential function, both of which exhibited minor influences on the strain rate sensitivity (Supplementary Fig. 11). These results show that the spall strength and the strain rate sensitivity of MGs are highly

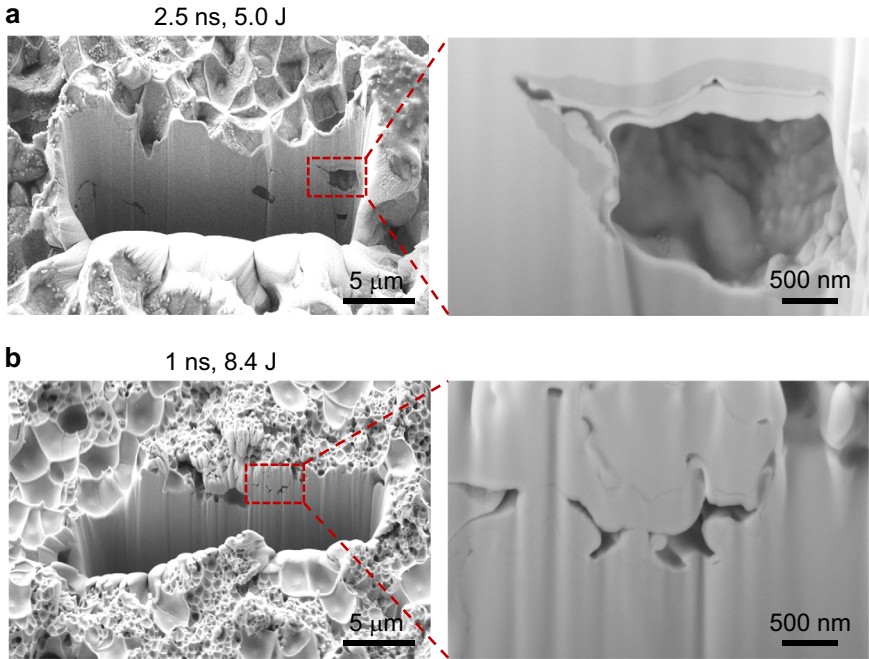

**Fig. 3 | Rate-dependent void distributions underneath the spall fracture surface.** SEM micrographs on the sidewall of the well milled by FIB show void growth and coalescence underneath the spall planes for the sample tested by the laser with 2.5 ns pulse duration and 5.0 J energy (**a**) and the laser with 1 ns pulse duration and 8.4 J energy (**b**).

sensitive to their internal structural defects. Furthermore, as the strain rate increased, the number of voids in MD simulations increased, but the average void size decreased (Fig. 2c, d and Supplementary Fig. 12). At relatively low strain rates, e.g., $1 \times 10^8 \, \text{s}^{-1}$, only a few isolated voids nucleated when the material reached its maximum stress. These voids grew independently as the material weakened and eventually failed. At high strain rates, e.g., $5 \times 10^9 \, \text{s}^{-1}$, however, the number of void nucleation sites increased sharply. The extremely close proximity of voids caused void growth and coalescence, which resulted in material failure.

Thus, our experiments and simulations helped us complete the failure mechanism diagram for metals shown in Supplementary Fig. 13. At extreme strain rates, the nucleation, growth, and coalescence of voids become the dominant failure mechanisms for MGs. However, the void growth kinetics are fundamentally distinct from those of crystalline metals, in which dislocation movement and twinning play crucial roles. To establish the connection between void growth and rate-dependent spall strength, we adopted the Curran-Seaman-Shockey model to describe the growth of voids under the control of surface energy[43]:

$$\frac{1}{D}\frac{dD}{dt} = \begin{cases} m(\sigma_h - \sigma_c), & \text{when } \sigma_h > \sigma_c \\ 0, & \text{when } \sigma_h \leq \sigma_c \end{cases} \quad (1)$$

where $m$ is a mobility coefficient, $\sigma_h$ is the hydrostatic stress, and $\sigma_c$ is the critical/threshold hydrostatic stress. Assuming that void instability followed the classical nucleation theory, then the critical stress $\sigma_c = 4\gamma/D$[44], in which $\gamma = 1.28 \, \text{J/m}^2$ is the surface energy obtained by MD simulations (see Methods). The hydrostatic stress in the uniaxial strain condition is $\sigma_h = (1+\nu)\sigma/[3(1-\nu)]$, where $\sigma = M\dot{\varepsilon}t$ is the normal stress in the thickness direction and $\nu = 0.41$ is the Poisson's ratio given by MD. Solving Eq. (1) yields the evolution of the void size (the details are given in the Supplementary Notes):

$$D(t) = \begin{cases} \left[D_0 \exp(-\chi^2 t_0^2) - \frac{2\sqrt{\pi}m\gamma}{\chi}(\text{erf}(\chi t) - \text{erf}(\chi t_0))\right]\exp(\chi^2 t^2), & t > t_0 \\ D_0, & t \leq t_0 \end{cases} \quad (2)$$

where $D_0$ is the initial void diameter, $\chi = \sqrt{\frac{M\dot{\varepsilon}m(1+\nu)}{6(1-\nu)}}$ and $t_0 = \frac{12\gamma(1-\nu)}{D_0 M\dot{\varepsilon}(1+\nu)}$. Setting the mobility coefficient $m = 25 \, (\text{Pa·s})^{-1}$ and $D_0 \sim 0.4 \, \text{nm}$, we noted remarkable agreement between the void evolution curves predicted by Eq. (2) and the void growth trends from MD simulations for all strain rates (Supplementary Fig. 14); these results validated our model.

Although the material strength increased with the strain rate in our MD simulations, we noted that the critical void diameter (when all curves reached their peak stresses) remained nearly constant ($D_c \approx 7.3 \, \text{nm}$), as shown in Supplementary Fig. 10d. This constant critical void diameter stemmed from the onset of mechanical instability due to competition between surface energy and strain energy; the details of the derivations are shown in the Supplementary Notes. Therefore, we used $D(t_c) = 7.3 \, \text{nm}$ as the criterion to estimate the spall strength ($\sigma_s = M\dot{\varepsilon}t_c$) for the specimens in the laser shock tests. When we employed a smaller mobility coefficient $m = 0.25 \, (\text{Pa·s})^{-1}$ and $D_0 \sim 2 \, \text{nm}$ (ref. 42) and kept all the parameters the same as those used in the MD results, our model predicted strain rate dependence of the spall strength that was in excellent agreement with our experiments ($1 \times 10^7 \, \text{s}^{-1} < \dot{\varepsilon} < 3 \times 10^7 \, \text{s}^{-1}$), as shown in Fig. 1b. Notably, mobility coefficient used for real materials was much lower than that used for MD simulations. This was due to the substantial difference between the energy states of the in-silico model and the real material. Wang et al. emphasized that the substantially faster cooling rates for in-silico MG models than those for practical MGs prepared by melt-spinning resulted in a far smaller activation energy for atomic motion and, thus, significantly greater mobility[45].

Even though the experiments by Coakley et al. and Righi et al. reached strain rates similar to those in this study[16,17], the tested polycrystalline or single-crystal metals exhibited spall strengths < $M/20$. This is because the plastic deformation of crystalline metals is mediated mainly by dislocation and twinning activities. Therefore, the void growth model proposed by Wilkerson and Ramesh[46,47] is suitable for

their cases:

$$\frac{1}{D}\frac{dD}{dt} = \begin{cases} \frac{1}{3}bn_m c_s \tanh\left[\frac{3}{4}\frac{b}{Bc_s}(\sigma_h - \sigma_c)\right], \text{when } \sigma_h > \sigma_c \\ 0, \text{when } \sigma_h \le \sigma_c \end{cases} \quad (3)$$

where $n_m$ is the mobile dislocation density, $b$ is the Burgers vector, $c_s$ is the shear wave speed and $B$ is the drag coefficient. Taking the parameters for copper, for instance, $n_m = 2 \times 10^{17}$ m$^{-2}$ (ref. [18]), $b = 0.25$ nm, $c_s = 2469$ m/s, and $B = 1.6 \times 10^{-5}$ Pa $\cdot$ s (ref. [48]), we estimated the void growth rate $\dot{D}$ from 40 to 400 m/s under an overpressure ($\sigma_h - \sigma_c$) of approximately between 1 and 4 GPa. This estimated rate was in good agreement with a recent experimental characterization of Cu (from 50 to 680 m/s)[16]. In the absence of dislocation-mediated mechanisms, the growth rate in Cu-Zr MG is only approximately from 0.3 to 12 m/s, nearly two orders of magnitude slower than in crystalline metals. This much lower void growth rate endows our material with exceptional spall strength. Figure 4 summarizes the state-of-the-art experimental measurements of the ultimate strength of various metallic materials[8–11,14–17,23,25–28,49–52]; our measurements advance the record for strength to ~M/13.

In summary, we conducted shock experiments on Cu-Zr MG at ultrahigh strain rates near the capacity of MD simulations by employing nanosecond laser pulses. Our study raised the measured strength of metallic materials to the unprecedented level of $M/13$ and approached the theoretical limit. Due to inadequate time for the development of shear banding at strain rates faster than $1.0 \times 10^7$ s$^{-1}$ (loading time-scale less than 5 ns) in our experiments, the material failed predominantly due to collective void nucleation and growth. Large-scale MD simulations and strain-rate-dependent hierarchical void structures on the spall surfaces indicated that faster strain rates activated a greater number of TTZs in MGs. When the voids reached a critical size, the material weakened due to mechanical instability from competition between surface energy and strain energy. We demonstrated that a void growth model governed by surface energy accurately depicted the strain rate dependence of the spall strength. In this study, the mechanical properties and failure mechanisms revealed by the extremely fast mechanical loading conditions enhanced our understanding of the time-dependent behavior of amorphous solids. Our findings also provided new prospects for utilizing amorphous phases to optimize the performance and design of metallic materials for applications under extremely fast mechanical conditions. Future research should

aim to explore the mechanical properties of other amorphous alloys at extreme strain rates to demonstrate the potential universality of the ultrahigh strength observed on Cu-Zr MGs herein. Further research should also concentrate on competition between shear banding and cavitation instability in MGs under extreme conditions, as cavitation has been largely overlooked in current plasticity theories for disordered materials[53].

## Methods

### Laser-induced shock experiments

Shock experiments were performed using the Shenguang-II Nd:glass laser facility (converted to a wavelength of 351 nm) at the National Laboratory for High Power Lasers and Physics in Shanghai, China (Supplementary Fig. 1a−c). The temporal profile of the laser pulse was approximately square, and pulse durations (full width at half maximum) of 1 ns and 2.5 ns were adopted (Supplementary Fig. 2). A lens array (LA) was used to eliminate the large-scale spatial modulation and obtain a flat-topped profile in the focal plane (Supplementary Fig. 1d, e)[54]. The optical system (lens + LA) had a focal spot over a flat region of dimensions ~0.5 × 0.5 mm$^2$. Laser energy in the range of 2–15 J was chosen so that spallation could occur without the laser completely plasmarizing the samples. Before each test, the sample was polished to ensure a surface roughness less than 30 nm and then fixed inside the target chamber by a holder (Supplementary Fig. 1f and g). Time-resolved FSV profiles of the shocked samples were measured with VISAR (Supplementary Fig. 1i). The time window and resolution of the VISAR system were 20 ns and 20 ps, respectively.

Laser shock experiments on the MG samples with a step were used to measure the longitudinal speed of sound (Supplementary Fig. 4a)[52]. The time delay $t_d$ between FSV signals for two surfaces was measured. In this way, the longitudinal speed of sound $c = h/t_d = 4487$ m/s was obtained; it was in good agreement with the value obtained from molecular dynamics simulations ($c = 4340$ m/s)[55]. The material density $\rho = 7.43$ g/cm$^3$ was measured using the Archimedes method, with the mass measured by electronic balance (XPR404S, Mettler Toledo) and the volume was measured by nano-CT (SkyScan 2214, Bruker). The obtained density $\rho = 7.43$ g/cm$^3$ was in good agreement with the values in the literature for Cu$_{50}$Zr$_{50}$ (7.62 g/cm$^3$ from measurements on bulk samples, and 7.30 g/cm$^3$ from MD simulations)[29,55]. The amplitude of the compressive stress of a shock wave was calculated as $\sigma_p = \rho c v_{fsp}/2$, where $v_{fsp}$ is the peak value of FSV. The estimated shock width ranged from 15 to 20 μm (determined based on the speed of sound and the

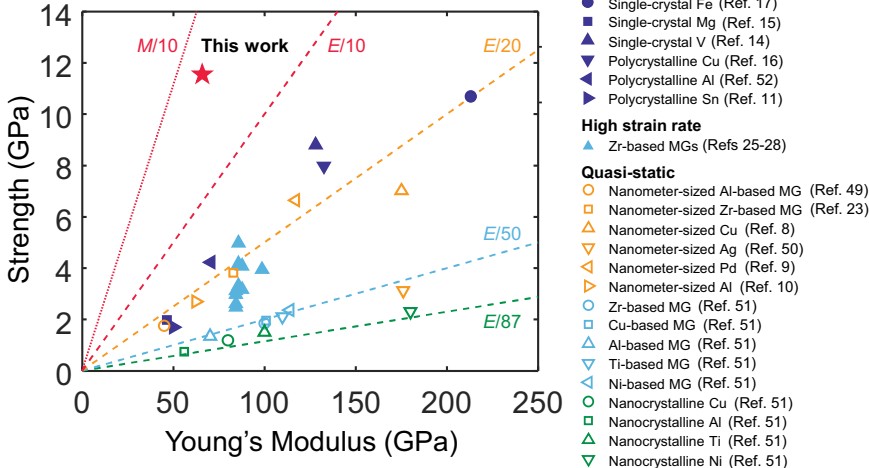

**Fig. 4 | Summary of the measured ultimate strengths for metallic materials.** Representative experimental results for crystalline and amorphous metals tested under quasi-static conditions[8–10,23,49–51], at high rates (10$^4$ s$^{-1}$ < $\dot{\varepsilon}$ < 10$^6$ s$^{-1}$)[25–28], and ultrahigh rates ($\dot{\varepsilon}$ > 10$^6$ s$^{-1}$)[11,14–17,52] are included for comparison. Our study on Cu-Zr MG shows the record-setting spall strength of ~M/13 for metals.

duration of the shock that lasted ~3 to 5 ns). The shock width is significantly larger than the local critical void diameter, $D_c$, by several orders of magnitude. The spall strength was calculated using $\sigma_s = \rho c \Delta v_{fs}/2$ based on the linear acoustic approximation[56], where $\Delta v_{fs}$ and $\Delta t$ are the velocity and time difference, respectively, as shown in Supplementary Fig. 4b. We also employed the formula with the thickness correction to calculate the spall strengths[57,58]. The thickness correction formula gave a slightly higher spall strengths, but the difference between two methods is less than 6.2% (see Supplementary Table 1). Therefore, we still report the results using the classic formula $\sigma_s = \rho c \Delta v_{fs}/2$ in Fig. 1b. The tensile strain rate was obtained by $\dot{\varepsilon} = \Delta v_{fs}/(2c\Delta t)$. The tensile strains imparted on the MG samples were estimated as $\varepsilon = \Delta v_{fs}/(2c)$, ranging from 4.5% to 7.7%.

## Sample preparation and post-mortem characterizations of the spall planes

$Cu_{50}Zr_{50}$ MG samples with in-plane dimensions of $2 \times 2$ mm$^2$ and thickness of approximately 50–100 μm were prepared for laser shock experiments using the single-roller melt spinning method. X-ray diffraction (Empyrean XRD, Malvern Panalytical Ltd) was performed to verify the amorphous state of the samples (Supplementary Fig. 6a and 9a). After laser shock tests, the MG samples were retrieved, and microscopy (Hitachi FE-SEM S4800) was used to characterize the plane where spalling occurred. Then, the void distribution underneath the spall plane was characterized using a focused ion beam (FIB) to mill a rectangular well in the sample (ZEISS Crossbeam 340). HRTEM and selected-area electron diffraction (SAED) characterizations were performed using Tecnai G2 F20 S-TWIN (FEI, US) at an accelerating voltage of 200 kV.

## Tensile tests

Uniaxial tensile tests at room temperature were conducted on a micro-tester (MT300, Deben) with a nominal strain rate of $1.5 \times 10^{-4}$ s$^{-1}$ under SEM (Phenom XL G2, Thermo Fisher Scientific). The $Cu_{50}Zr_{50}$ ribbon sample with a cross-section of $2.3 \times 0.053$ mm$^2$ was used. The tensile strain was obtained using the open-source digital image correlation MATLAB code, Ncorr[59]. The Young's modulus was determined to be $65.9 \pm 2.6$ GPa from the stress-strain curves (Supplementary Fig. 15).

## Molecular dynamics simulations

Molecular dynamics simulations were conducted to explore the cavitation kinetics of $Cu_{50}Zr_{50}$ MG under ultrahigh strain rates. The simulations were performed using LAMMPS open source code[60]; an embedded-atom method (EAM) interatomic potential developed by Mendelev et al. was employed[61]. In all the simulations, periodic boundary conditions were applied to all three axes. First, the simulation box containing 16 million randomly distributed Cu atoms and 16 million Zr atoms was heated to 2000 K and held for 200 ps. Then, the system was cooled to 300 K at a rate of $1.7 \times 10^{12}$ K s$^{-1}$. After that, the system equilibrated at 300 K for 200 ps. In all these processes, an isothermal-isobaric (NPT) ensemble was employed, and a time step of 1 fs was used. Finally, this simulation gave an atomistic model for $Cu_{50}Zr_{50}$ MG with dimensions of approximately $80 \times 80 \times 80$ nm$^3$. The glassy state of the model was characterized by the pair distribution function shown in Supplementary Fig. 10b. The glass transition temperature was approximately 750 K and determined by the change in the slopes of the potential energy vs. temperature curve during quenching[62].

Then, uniaxial strain tests were performed. The model was stretched along the $x$-axis to 20% strain at nominal strain rates ranging from $1 \times 10^8$ s$^{-1}$ to $5 \times 10^9$ s$^{-1}$. Meanwhile, the dimensions of the box along the $y$-axis and $z$-axis were fixed to ensure that the model deformed under uniaxial strain conditions. The microcanonical (NVE) ensemble was employed, and a time step of 1 fs was used in the mechanical tests.

To calculate the surface energy of $Cu_{50}Zr_{50}$ MG, two quenched samples were equilibrated at 300 K with a time step of 1 fs using NPT ensembles. One of the systems had periodic boundary conditions along all three axes, while the other system had periodic boundary conditions in the y- and z-directions and free surface boundary conditions in the x-direction. The total potential energy of the two systems averaged over 20 ps was used in the surface energy calculation.

To check the effect of the cooling rate on the strain rate sensitivity, three smaller ($20 \times 20 \times 20$ nm$^3$) $Cu_{50}Zr_{50}$ MG models at different cooling rates ($1.7 \times 10^{12}$, $6.8 \times 10^{11}$ and $1.7 \times 10^{11}$ K s$^{-1}$) were prepared (Supplementary Fig. 11a). We then used a logarithmic function to fit the relation between the energy per atom at 300 K and the cooling rate. Next, we performed hybrid MD/Monte Carlo (MC) simulations in the variance constrained semi-grand canonical ensemble, following the method in ref. 41. The energies per atom at 300K obtained from the hybrid MD/MC simulations were used to determine the corresponding effective cooling rates by extrapolating the logarithmic relationship obtained from the direct MD simulations. This way, we obtained four $Cu_{50}Zr_{50}$ MG models ($20 \times 20 \times 20$ nm$^3$) at much lower effective cooling rates ($2.1 \times 10^{11}$ K s$^{-1}$, $2.0 \times 10^{10}$ K s$^{-1}$, $8.6 \times 10^9$ K s$^{-1}$ and $1.1 \times 10^7$ K s$^{-1}$). Finally, we performed uniaxial strain tensile tests on the four MG models at two strain rates ($5.0 \times 10^8$ s$^{-1}$ and $5.0 \times 10^9$ s$^{-1}$) using the embedded-atom method (EAM) interatomic potential developed by Mendelev et al.[61]. The results show a strain rate sensitivity that is nearly identical to that of the larger model ($80 \times 80 \times 80$ nm$^3$) which was cooled at $1.7 \times 10^{12}$ K s$^{-1}$ (Supplementary Fig. 11b).

To discuss whether different interatomic potential functions would affect the strain rate sensitivity, we performed additional uniaxial strain tensile tests on the smaller $Cu_{50}Zr_{50}$ model ($20 \times 20 \times 20$ nm$^3$, $1.7 \times 10^{12}$ K s$^{-1}$) using two EAM potential functions[61,63] and a modified embedded-atom method (MEAM) potential function[64]. Two tensile strain rates ($5 \times 10^8$ s$^{-1}$ and $5 \times 10^9$ s$^{-1}$) were applied. Supplementary Fig. 11c and d demonstrate that several potential functions had similar strain rate sensitivities, despite variations in elastic moduli and material strengths.

## Data availability

All data are available in the main text or the supplementary materials. The statistical void size data and strength data generated in this study are provided in the Source Data file. Additional data related to this paper may be requested from the authors.

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

## Acknowledgements
The experiment was performed using Shenguang-II Nd:glass high-power laser facility at the National Laboratory on High Power Laser and Physics in Shanghai, China. X.W. greatly appreciates the support by the National Natural Science Foundation of China (Grant Nos. 12325202, 12172005, 11890681, and 11988102), and the National Key R&D Program of China (Grant No. 2022YFB3806102). H.G. and Z.L. acknowledge a research startup grant (002479-00001) from Nanyang Technological University and the Agency for Science, Technology and Research (A*STAR) in Singapore.

## Author contributions
W.Z. and H.S. both carried out the experiments and analyzed the results. W.Z. and Z.L. developed and validated the theoretical model. Z.L. performed the MD simulations and data analyses. X.W. and H.G. conceived the concept and supervised the study. All authors contributed to manuscript writing.

## Competing interests
The authors declare no competing interests.
