## [Peer Review File · Nature Communications]

Amorphous alloys surpass E/10 strength limit at extreme strain ratesReviewers' comments:

Reviewer #1 (Remarks to the Author):

In this manuscript, the authors investigate the ability of a metallic glass, CuZr, to surpass E/10 strength limit at extreme strain rates. The manuscript is well written and concise with its findings. While the work is an interesting area of research, there are several questions/concerns the reviewer has regarding various aspects of the work. The greatest of those concerns being more microstructural proof is needed before the authors can determine the failure mechanism at play in amorphous metals/metallic glasses when tested at ultrahigh strain rates. Additionally, the extremely high spall strength reported (9.8 GPa) causes the reviewer to question the validity of the laser shock setup. Consequently, the reviewer requires major revision before considering the manuscript for publication. These specific questions/concerns are listed below.

The first issue with the manuscript is not related to the scientific merit, but rather the listing and numbering of the Figures throughout the manuscript. All reference to the figure within the main text are misnumbered/mislabeled. For example, the second sentence of the results section the authors discuss using a nanosecond Nd:glass laser to generate a shock and reference Figure 2a. When viewing Figure 2a, the reviewer sees the spall surfaces of the sample experiencing different input laser energies. This error seemed to occur every time a Figure (that was included within the manuscript and not the supplementary) was referenced. The second issue regarding the labeling of figures was the order in which the supplementary figures appeared in the manuscript. Their order was random and not sequential as seen by the first supplemental figure discussed in the manuscript being Supplementary Figure 6. These corrections will improve the readability of the manuscript.

Scientific Questions/Concerns

Can the authors comment on why a 50:50 atomic composition was selected for their metallic glass? The authors wrote, "MGs consist of stable regions where atoms are more densely packed and rheological regions where atoms are more loosely packed, i.e. defective spots." Are these stable regions with more densely packed atom not considered crystalline?

Did all the specimen fully spall? Or did any of lower strain rate test cause incipient spall within the sample? If some of the sample did undergo incipient spall, can the authors provide optical or SEM images of the specimens' cross section(s)? This evidence would further support the idea of void growth and coalescence as the main failure mechanism in amorphous metals/metallic glasses as the authors propose in Supplementary Figure 8.

Additionally, the authors wrote, "However, the strain rate sensitivity of the material strength obtained from MD simulations was notably less than that found in our experiments, since the material strength approached its limit at M/10." This statement is interesting for a variety of reasons. Can the authors explain why the MD simulations showed less strain rate sensitivity than the actual experiment? Additionally, the material showing an increase in ultimate strength with increasing strain rate is consistent with the behavior of a metallic glass behaving as a non-Newtonian fluid. With that said, did the authors perform any XRD or selected area diffraction of material post spall testing? The reviewer would be interested to know if the high stress imparted on the MG during the spall testing caused crystallization of the sample. Also, the authors provide the XRD spectrum with a major and minor peak present for the as-cast MG, which hints at the material not being fully amorphous. Thus, did they perform TEM on the as-cast condition to determine if any nanoscale crystallites were present? This microstructural characterization is critical before any conclusions can be clearly drawn regarding the failure mechanism at play in the metallic glass since the presence of the nanocrystallites would result in shear localization. Finally, the interface between the possible nanoscale crystallites and the surrounding amorphous material could also be a point of weakness causing failure within the material. The authors report the spall strength of the metallic glass being 9.8 GPa, which is astronomically high value when considering the spall strength for single crystal Cu is 4.5 GPa. (The reviewer was unable to find a value for single crystal Zr.) This causes the reviewer to question the validity of the laser induced shock test setup. Have the authors attempted the same experiment using gas gun systems?

Finally, the authors report the density of their metallic glass as 6.30 g/cm³. The reviewer understands a metallic glass having a lower density due to the inefficient packing of atoms. However, the material is composed of 50 atomic percent Cu and Zr, and their densities are 8.96 and 6.49 g/cm³ respectively. Consequently, the reviewer does not understand how the reported density can be lower than the element with the lower density of a binary system. Please explain.

Reviewer #2 (Remarks to the Author):

This paper presents molecular dynamic and laser shock experiments on Cu-Zr bulk metallic glass, estimating an extreme 'strength' limit at 10^7 1/s. While the work appears as if it could have merit, the paper in present form makes it difficult to ascertain if the extreme values presented are fully validated. Some of the concerns are as follows:

-There are no details on the experimental setup, or material preparation work performed to the samples for the laser shock experiments. It would be useful to provide that in the supplemental documents.

- How are you calculating the Poisson ratio of ~ 0.4 you mention on page 3? This can be tricky given the material of interest is being hit with a high-energy laser, yet it appears you are basing your 1D estimations of "strength" as a function of Young's modulus from it, and it may be quite sensitive to it.

- You are essentially reaching the Debye frequency at these extreme strain rates. There has been more work in laser spall in manuscripts than is described in the paper. The reviewer suggests the authors take another look at literature. For example, the authors reference Righi's work, but it seems they are suggesting they are the first to find these high values, when papers like this show the same type of response on a Zr-Cr BMG - although not quite as directly stated. It's unclear what is significantly improved from this suggested work to the authors work - rather than incremental.

- Fig. 1b, the MD results do not appear to trend as the data. This is concerning. Why the slope change? What is not being captured (physics-based) in the simulations to account for this change in slope? Also, the model shows a trend that the data (from references) at lower strain-rates does not seem to support. How do you explain this?

-It is unclear who's data is who's in Fig.5. The references are in the description, but not listed on the plot itself, so the reader cannot figure out where the data is coming from explicitly.

Reviewer #3 (Remarks to the Author):

This manuscript has some interesting data, but it is not clear that the authors' conclusions are justified.

This reviewer has serious concerns about the analysis of the free surface velocity data in SFig1 that are at the core of the paper. The pulse durations are stated as 1 ns and 2.5 ns. However, the only data presented is the FSV obtained after passage through the samples. The authors' earlier paper on the technique (Ref. 43) claimed that the rise time of the laser pulse was 300 ps. Assuming that the durations they state are the FWHM values, that should provide about 600-700 ps of a constant power. The question for a true spall test, however, is whether a steady shock has actually been formed in the specimen, with a compressed state for some time before the release begins. It is not clear that such a steady shock is ever formed in the specimen in this setup with either the 1 ns or the 2.5 ns pulse. Certainly the free surface velocities do not show such an initial shock state.

As a consequence, the authors' computation of the pullback velocity and the spall strength is not correct, more likely representing a more complex stress state with a much wider range of transients. This also leads to an unreasonable estimate of the tensile strain rate, and a lack of knowledge of the

prior shock stress. It is just as likely that the purported "spall strength" increase with strain rate shown in fig. 1 b is driven by the change in the pulse amplitude (i.e., the prior non-steady shock stress) rather than the poorly calibrated strain rate. Note that the authors should also be computing the corrected strain rate.

Finally, the authors conflate the spall strength - which is a mechanistic quantity, not a material property - with the ideal strength. In most metals, the spall strength is a multiple of the yield strength, sometimes 3-5 times the yield strength. The purported unusual behavior here is not sensible in this light.

Response to the editor and reviewers

We thank the editor and reviewers very much for their constructive comments. Changes have been made in the revised manuscript (text highlighted in yellow) based on reviewers' comments. All page numbers in our responses refer to the revised manuscript unless otherwise stated. The following are the itemized responses to the reviewers' comments.

Comments from the editors and reviewers:

Reviewer #1

The first issue with the manuscript is not related to the scientific merit, but rather the listing and numbering of the Figures throughout the manuscript. All reference to the figure within the main text are misnumbered/mislabeled.

- We have double-checked and corrected all the references to the figure in the main text.*

Scientific Questions/Concerns

Can the authors comment on why a 50:50 atomic composition was selected for their metallic glass?

- We thank the referee for this question. As shown by Tang et al. [1], $\text{Cu}_{50}\text{Zr}_{50}$ has the best glass formation ability among $\text{Cu}_x\text{Zr}_{1-x}$ binary alloy for $20 \leq x \leq 80$ at.%. Since then, many experimental and theoretical studies have used $\text{Cu}_{50}\text{Zr}_{50}$ as the model material to study the properties of metallic glasses. Thus, we chose 50:50 atomic composition for our CuZr metallic glass to make sure the material is in a fully amorphous state.*

The authors wrote, “MGs consist of stable regions where atoms are more densely packed and rheological regions where atoms are more loosely packed, i.e. defective spots.” Are these stable regions with more densely packed atom not considered crystalline?

- We thank the referee for bringing up this important question. Metallic glasses are fully amorphous, but can also have intrinsic structural heterogeneities with atoms in some regions (in the order of several cubic nanometers) more densely packed than the other regions. Nonetheless, the material density in the densely packed region is still notably less than that in the crystalline phase. The intrinsic structural heterogeneity in metallic glasses has been demonstrated by Zhu et al. [2] using atomic-resolution dark-field transmission electron microscopy (TEM). The contrast between the dark and bright regions (corresponding to the densely packed and loosely packed regions) in the images of Zr-Cu-based metallic glass taken under HAADF-STEM mode clearly demonstrated such intrinsic structural heterogeneity.*

Did all the specimen fully spall? Or did any of lower strain rate test cause incipient spall within the sample? If some of the sample did undergo incipient spall, can the authors provide optical or SEM images of the specimens' cross section(s)? This evidence would further support the idea of void growth and coalescence as the main failure mechanism in amorphous metals/metallic glasses as the

authors propose in Supplementary Figure 8.

- *We thank the referee for the very constructive suggestion. We have one specimen that was tested by a less intense laser pulse (1.3J energy and 1ns duration). The specimen did not fully spall. We used focus-ion-beam (FIB) to cut into the specimen from the back surface. At a depth of approximately 5 microns, we did discover features of nanovoids nucleation and coalescence. Thus, this observation together with previous characterization of the spalled specimens supports our conclusion that cavitation is the main failure mechanism for metallic glasses under ultra-high strain rates.*
- *We have added Supplementary Fig. 8 and discussions on Page 3 to elucidate the findings on the specimen tested under lower strain rate:*

To further confirm cavitation is the main failure mechanism in our tests, we milled into an un-spalled specimen tested by a less intense laser pulse (1.3J energy and 1 ns duration) using FIB. At a depth of approximately 5 microns from the back surface, we discovered features of cavity initiation and coalescence (Supplementary Fig. 8).

Additionally, the authors wrote, “However, the strain rate sensitivity of the material strength obtained from MD simulations was notably less than that found in our experiments, since the material strength approached its limit at M/10.” This statement is interesting for a variety of reasons. Can the authors explain why the MD simulations showed less strain rate sensitivity than the actual experiment?

- *We thank the referee for raising this important question. There could be several causes for the notable difference between the strain rate sensitivities obtained from MD simulations and from experiments. For instance, there is still a big difference in size between the MD model (80 nm × 80 nm × 80 nm) and the real specimens (100 μm × 2 mm × 2 mm). The cooling rate by melt spinning method for the real specimens was in the order of 10⁴-10⁶ K/s, much lower than that used in MD model (1.7×10¹² K/s). This could result in a difference in the structural heterogeneity between MD model and the real specimens. The most likely factor is that the real specimens usually contained scattered “defects” such as nanopores, while the MD model was relatively “ideal.” To test this hypothesis, we created two “defective” atomistic models by removing atoms in the center of the original MD model. One defective model contained a nanovoid with a diameter of 1.4 nm, and the other contained a nanovoid with a diameter of 4 nm. Applying uniaxial strain tests on these two atomistic models with defects, we obtained strain rate sensitivities that were notably larger than the “ideal” MD model. The model having a small defect yielded a strain rate sensitivity of 0.16, while the model having a larger defect yielded a strain rate sensitivity of 0.22, getting close to the value of 0.52 obtained from experiments.*
- *We have modified Fig. 1b and added more discussions on Page 4 to elucidate the possible causes for the difference in the strain rate sensitivity between the MD calculations and experiments:*

This could be attributed to the fact that the atomistic model was substantially smaller than the real specimens. In addition, the extreme cooling rate used for generating the atomistic model (1.7×10¹² K/s) was significantly faster than those from melt spinning (10⁴-10⁶ K/s), which could result in different nonequilibrium states between the atomistic model and real specimens. For instance, the real specimens might contain scattered “defects” such as nanopores, while the MD model is rather “ideal”. To investigate this effect, we generated two “defective” atomistic models that contained

nanovoids with diameters of 1.4 nm and 4 nm, respectively, by removing some atoms in the center. Applying uniaxial strain tensile tests on these two models revealed that the strain rate sensitivity was highly sensitive to the flaw size – the model with a smaller nanopore showed a strain rate sensitivity of 0.16, while the one with a larger nanopore showed a strain rate sensitivity of 0.22, both of which were significantly greater than the ideal model (Fig. 1b). These results show that the spall strength of MGs is highly sensitive to their internal structural states.

Additionally, the material showing an increase in ultimate strength with increasing strain rate is consistent with the behavior of a metallic glass behaving as a non-Newtonian fluid. With that said, did the authors perform any XRD or selected area diffraction of material post spall testing? The reviewer would be interested to know if the high stress imparted on the MG during the spall testing caused crystallization of the sample.

- *This is a very inspiring comment. We have performed X-ray diffraction (XRD) and selected-area electron diffraction (SAED) on the spalled samples. XRD patterns of the two spalled specimens showed the same broadened diffuse humps as in the as-cast specimens. SAED results on different locations below the spall plane also only showed diffuse halos, which were consistent with the disordered atomic structures observed in high resolution TEM (HRTEM) images. Therefore, we believe that the MG did not crystallize during or after the laser shock tests.*
- *We have included XRD and SAED characterizations in Supplementary Fig. 9, and added more discussions on Page 3:*

Last, X-ray diffraction (XRD) and selected-area electron diffraction (SAED) in a transmission electron microscope (TEM) were carried out to assure that the amorphous nature was maintained in the specimens after tests. XRD spectra of the two post-test specimens revealed the same broadened diffuse humps as in the as-cast specimens. SAED on different spots below the spall plane also only showed diffuse halos, consistent with the disordered atomic structures observed in high-resolution TEM (HRTEM) images (Supplementary Fig. 9).

Also, the authors provide the XRD spectrum with a major and minor peak present for the as-cast MG, which hints at the material not being fully amorphous.

- *We have checked that the XRD spectrum for the as-cast MG in this study is consistent with the XRD spectrum of Zr-Cu-based MG in the literature [3, 4]. The major and minor “peak” in the XRD spectrum mentioned by the referee corresponds to a series of broad diffraction maxima with FWHM ~ 10 degrees manifesting a fully amorphous structure, in comparison with sharp Bragg peaks in crystalline materials. In the field of amorphous alloys, they are often referred as “broadened diffuse humps” which is a typical feature of the amorphous phase.*

Thus, did they perform TEM on the as-cast condition to determine if any nanoscale crystallites were present? This microstructural characterization is critical before any conclusions can be clearly drawn regarding the failure mechanism at play in the metallic glass since the presence of the nanocrystallites would result in shear localization. Finally, the interface between the possible nanoscale crystallites and the surrounding amorphous material could also be a point of weakness causing failure within the material.

- *We thank the referee for bringing up this concern. We have performed TEM characterization on MG sample on the as-cast condition and confirmed that the sample was in a fully amorphous state (see Supplementary Fig. 6b).*
- *Combining the characterizations after the shock tests, we thus are confident that cavitation is the main failure mechanism for CuZr MG in our study, and have emphasized our finding on Page 4: The above characterizations indicated that no crystallization or related crystal plasticity mechanism happened during or after the tests, and MG failed predominantly due to void growth and coalescence rather than shear banding. The main factor differing our tests from previous ones is the ultra-high strain rate.*

The authors report the spall strength of the metallic glass being 9.8 GPa, which is astronomically high value when considering the spall strength for single crystal Cu is 4.5 GPa. (The reviewer was unable to find a value for single crystal Zr.) This causes the reviewer to question the validity of the laser induced shock test setup. Have the authors attempted the same experiment using gas gun systems?

- *We thank the referee for the suggestion. We did not attempt the same experiments using a gas gun system which could only reach strain rates in the range of 10^4 to 10^6 s^{-1} [5, 6], far slower than the extreme strain rates in this study. The results on similar Zr-based MGs using gas gun systems from the literature were included in Fig. 1b for comparison.*
- *Second, although no result on single crystal Cu tested at comparable strain rates had been reported, we noted that the spall strength of polycrystalline Cu reached as high as 8.5 GPa at strain rates of 0.5×10^9 s^{-1} , as reported by Coakley et al. who employed picosecond laser pulse [7]. Such high spall strength reaches $M/25$ in which M is the pressure wave (P -wave) modulus. Nonetheless, the normalized spall strength of polycrystalline Cu is still lower than that of our CuZr MG ($\sim M/13$), primarily because (as we discussed in the Introduction) plastic deformation mechanisms such as dislocations and twinning in single-crystal and nanocrystalline metals could reduce material strength even at such extreme strain rates.*
- *Therefore, given that laser shock-induced spallation is a mature technique and has been widely employed to study the mechanical properties of metals over the past 10 years, we are confident that the results obtained in our study are reliable.*

Finally, the authors report the density of their metallic glass as 6.30 g/cm³. The reviewer understands a metallic glass having a lower density due to the inefficient packing of atoms. However, the material

is composed of 50 atomic percent Cu and Zr, and their densities are 8.96 and 6.49 g/cm³ respectively. Consequently, the reviewer does not understand how the reported density can be lower than the element with the lower density of a binary system. Please explain.

- *We are grateful to the referee for pointing out this issue. We examined the material density measurement and found that the biggest error came from the uncertainty in the sample volume. In the revision, we improved the measurements by using high resolution nano-CT (SkyScan 2214, Bruker) to measure the volume of the Cu₅₀Zr₅₀ MG ribbon. The corrected material density is 7.43 g/cm³, comparable with the previously reported value of bulk Cu₅₀Zr₅₀ specimens (7.62 g/cm³) and that obtained from MD simulations (7.30 g/cm³) [8, 9].*
- *The corrected material density brought the maximum spall strength up to 11.5 GPa. However, it did not change the normalized spall strength as σ_s and E are both proportional to the density. Therefore, the peak normalized spall strength σ_s/E in our study remains approximately 1/6.*

Reviewer #2

This paper presents molecular dynamic and laser shock experiments on Cu-Zr bulk metallic glass, estimating an extreme 'strength' limit at 10⁷ 1/s. While the work appears as if it could have merit, the paper in present form makes it difficult to ascertain if the extreme values presented are fully validated. Some of the concerns are as follows:

-There are no details on the experimental setup, or material preparation work performed to the samples for the laser shock experiments. It would be useful to provide that in the supplemental documents.

- *We thank the referee for this suggestion. In this revision, we have provided more details about tests in the Methods on Page 6 and added Supplementary Fig. 1 to show the experimental setup.*

- How are you calculating the Poisson ratio of ~0.4 you mention on page 3? This can be tricky given the material of interest is being hit with a high-energy laser, yet it appears you are basing your 1D estimations of "strength" as a function of Young's modulus from it, and it may be quite sensitive to it.

- *We thank the referee for bringing up this important point. We used the Poisson's ratio $\nu=0.4$ for Cu₅₀Zr₅₀ MG for the following reasons. First, previous study gave $\nu=0.39$ for bulk CuZr MG (with the atomic ratio close to 1:1) [9]. Second, the Poisson's ratio can be calculated using $\nu = (\sigma_y + \sigma_z)/(2\sigma_x + \sigma_y + \sigma_z)$, in which σ_x , σ_y , and σ_z are the three normal stresses obtained from MD simulations of the uniaxial strain tensile test on our atomistic Cu₅₀Zr₅₀ model (x is the loading direction). This approach led to $\nu=0.41$. In fact, although the energy state of atomistic MG model generated by MD simulations might differ from that of the bulk MG, the change in the Poisson's ratio is only approximately 2.5%. Therefore, here we used $\nu=0.4$ for Cu₅₀Zr₅₀ MG and believed that the results should not be sensitive to the value of the Poisson's ratio.*

- You are essentially reaching the Debye frequency at these extreme strain rates. There has been more work in laser spall in manuscripts than is described in the paper. The reviewer suggests the authors take another look at literature. For example, the authors reference Righi's work, but it seems they are suggesting they are the first to find these high values, when papers like this shows the same type of response on a Zr-Cu BMG - although not quite as directly stated. It's unclear what is significantly improved from this suggested work to the authors work - rather than incremental.

- *We thank the referee and have added more representative references about the laser induced spallation of metallic materials in the Introduction, and modified Figure 4 in the main text accordingly.*
- *Nevertheless, we believe there was a misunderstanding about our statement of impact of the spall strength of MG measured in this study. Although previous studies on the laser shock experiments on various single-crystal and polycrystalline metals (including the work by Righi et al.) have reported absolute spall strengths higher than the spall strength values reported in this study. Their normalized strengths (by Young's modulus E for uniaxial tensile tests or by pressure wave modulus M for uniaxial strain tests) have never surpassed $1/25$. Furthermore, most of the previous laser shock experiments were performed on single-crystal or polycrystalline metals. Studies on the mechanical properties of metallic glasses at such extreme strain rates are still lacking. In this study, we discovered that the normalized spall strength of $\text{Cu}_{50}\text{Zr}_{50}$ could reach $1/13$ (normalized by M) or $1/6$ (if normalized by E). This is the first time, to the authors' best of knowledge, that the strength of metallic materials has reached this level. Therefore, we are confident that our work has sufficient value and impact on the communities of materials science and mechanics.*

- Fig. 1b, the MD results do not appear to trend as the data. This is concerning. Why the slope change? What is not being captured (physics-based) in the simulations to account for this change in slope? Also, the model shows a trend that the data (from references) at lower strain-rates does not seem to support. How do you explain this?

- *We thank the referee for these inspiring questions that are partially related to one of the comments by the first referee. There could be several causes for the notable difference between the strain rate sensitivities obtained from MD simulations and from experiments. For instance, there is still a big difference in size between the MD model ($80 \text{ nm} \times 80 \text{ nm} \times 80 \text{ nm}$) and the real specimens ($100 \mu\text{m} \times 2 \text{ mm} \times 2 \text{ mm}$). The cooling rate by melt spinning method for the real specimens was in the order of 10^4 - 10^6 K/s, much lower than that used in MD model (1.7×10^{12} K/s). This could result in a difference in the structural heterogeneity between MD model and the real specimens. The most likely factor is that the real specimens usually contained scattered "defects" such as nanopores, while the MD model was highly "ideal." To test this hypothesis, we created two "defective" atomistic models by removing some atoms in the center of the original MD model. One defective model contained a nanovoid with a diameter of 1.4 nm, and the other contained a nanovoid with a diameter of 4 nm. Applying uniaxial strain tests on these two atomistic models with defects, we obtained strain rate sensitivities that were notably larger than the "ideal" MD model. The model having a small defect yielded a strain rate sensitivity of 0.16, while the model having a larger defect yielded a strain rate sensitivity of 0.22, getting close to the value of 0.51 obtained from experiments.*

- *For the referred data from the literature at lower strain rates, the predominant failure mechanisms for MGs differ from cavitation as discovered in this study. As reported by Tang et al., MGs tested at strain rates of $\sim 10^6 \text{ s}^{-1}$ failed due to the combination of cavitation (activation of tension transformation zones) and shear banding (activation of shear transformation zones) [10]. The change in the failure mechanisms would lead to a different trend of the spall strength at lower strain rates. Since our model assumed that the material failed mainly due to cavitation, the model should not apply to the lower strain rate regime.*
- *We have modified Fig. 1b and added more discussions on Page 4 to elucidate the possible causes for the difference in the strain rate sensitivity between the MD calculations and experiments:*
This could be attributed to the fact that the atomistic model was still substantially smaller than the real specimens. In addition, the extreme cooling rate used for generating the atomistic model ($1.7 \times 10^{12} \text{ K/s}$) was significantly faster than those in melt spinning (10^4 - 10^6 K/s), which could result in different nonequilibrium states between atomistic model and real specimens. For instance, the real specimens might contain scattered “defects” such as nanopores, while the MD model is rather “ideal”. To test this hypothesis, we generated two “defective” atomistic models that contained nanovoids with diameters of 1.4 nm and 4 nm, respectively, by removing some atoms in the center. Applying uniaxial strain tensile tests on these two models revealed that the strain rate sensitivity was highly sensitive to the flaw size – the model with a smaller nanopore showed a strain rate sensitivity of 0.16, while the one with a larger nanopore showed a strain rate sensitivity of 0.22, both of which were significantly greater than the ideal model (Fig. 1b). These results show that the spall strength of MGs is highly sensitive to their internal structural states.

-It is unclear who's data is who's in Fig.5. The references are in the description, but not listed on the plot itself, so the reader cannot figure out where the data is coming from explicitly.

- *In the revision, we have added the references to the legend in the figure (now Fig. 4).*

Reviewer #3

This manuscript has some interesting data, but it is not clear that the authors' conclusions are justified.

This reviewer has serious concerns about the analysis of the free surface velocity data in SFig1 that are at the core of the paper. The pulse durations are stated as 1 ns and 2.5 ns. However, the only data presented is the FSV obtained after passage through the samples.

- *We thank the referee for raising this important point. In the revision, we have added the laser profiles for 1ns and 2.5ns durations in Supplementary Fig. 2. How the free surface velocity was measured using the VISAR system was also illustrated in Supplementary Fig. 1.*

The authors' earlier paper on the technique (Ref. 43) claimed that the rise time of the laser pulse was 300 ps. Assuming that the durations they state are the FWHM values, that should provide about 600-700 ps of a constant power. The question for a true spall test, however, is whether a steady shock has actually been formed in the specimen, with a compressed state for some time before the release begins.

It is not clear that such a steady shock is ever formed in the specimen in this setup with either the 1 ns or the 2.5 ns pulse. Certainly the free surface velocities do not show such an initial shock state.

As a consequence, the authors' computation of the pullback velocity and the spall strength is not correct, more likely representing a more complex stress state with a much wider range of transients.

This also leads to an unreasonable estimate of the tensile strain rate, and a lack of knowledge of the prior shock stress. It is just as likely that the purported "spall strength" increase with strain rate shown in fig. 1 b is driven by the change in the pulse amplitude (i.e., the prior non-steady shock stress) rather than the poorly calibrated strain rate. Note that the authors should also be computing the corrected strain rate.

- *We would like to respond to the above three comments together, as they are all about the data analysis and result interpretation of the laser shock experiments.*
- *There are three main experimental methods to study dynamic fracture/spall of materials: (1) by plate impact, (2) by high explosives, and (3) by laser-induced shock waves. The plate impact method is known to have flat-top (or rectangular) pressure pulse and strain rate history. The latter two methods generate, to a good approximation, triangular shock waves (also known as the "Taylor wave" after G.I. Taylor), as shown in Fig. R1a. The laser induced spallation technique dates back to the 1960s [11]. After decades of development, it has become a mature method to study the dynamic mechanical properties of materials, for example, in the representative studies cited in the Introduction of the main text. The method of analyzing the wave propagation and interactions in the laser induced shock tests has been well summarized in the famous book, Spall Fracture, written by Antoun et al. [12]. As shown in Figs. R1b and C, the spalling stress is determined by the intersection of Riemann trajectories passing through the points $(\sigma = 0, v = v_b)$ for C_+ characteristic and $(\sigma = 0, v = v_a)$ for C_- characteristic. In the distance–time diagram of Fig. R1b, a family of the characteristic lines C_+ represent the right-going wave propagating towards the free surface. The shock front trajectory is described by the line $0a$. When the shock front reaches the free surface at point a , the free-surface velocity undergoes a jump from zero up to u_a . The unloading wave behind the shock front causes decay in the free-surface velocity as shown in Fig. R1d. Thus, before spall fracture, the particle velocity history has the same triangular shape as the stress history (Fig. R1a). At the free surface, the shock wave is reflected as a left-going wave that travels toward the interior of the sample. This rarefaction wave is represented by the characteristic line C_- emanating from point a . The state of the particles must satisfy conditions on both the C_+ and C_- characteristics and is determined in the stress–particle velocity diagram (Fig. R1c) by the intersection of Riemann trajectories describing states of matter along the C_+ and C_- characteristics that pass through the particle at any given time. Therefore, the peak tensile stress at the spall plane just before fracture corresponds to the intersection of trajectories as and $2s$. Line as describes the change of state along the tail C_- characteristic; line $2s$ represents the trajectory of the change of state along the last of the C_+ characteristics of the incident wave crossing the spall plane before the fracture. It can be*

seen from the stress–particle velocity diagram that the peak tensile stress (spall strength) is related to the pullback velocity, which is defined as the difference between the rear surface velocity maximum and first minimum peak, i.e., $\Delta v_{fs} = v_a - v_b$ (see Fig. R1d). According to the original work by Novikov in 1966 [13], the simplified acoustic approach yields the famous relationship between spall strength and the pullback velocity: $\sigma_s = \rho c \Delta v_{fs} / 2$, with c being the sound speed. Similarly, the strain rate can be estimated by $\dot{\epsilon} = \Delta v_{fs} / (2c\Delta t)$ [14]. The above methods for calculating the spall strength and strain rate have been widely used [15-17].

- The referee is correct that the laser induced spallation technique cannot produce a relatively steady shock within a finite width in the specimen as plate impact technique. Rather, the laser shock generates an ultrahigh tensile stress on a thin layer (i.e., the spall plane) of the specimen. However, this does not affect its applicability to investigate the fracture strength of materials at ultra-high strain rates. Therefore, we are confident with the data analysis and results interpretation in this study.

Fig. R1 Wave interaction diagrams for calculation of spall strength that is adopted from Ref [12]. a, Input compressive stress history. b, Distance-time diagram showing trajectories of loading and unloading waves. c, Stress-particle velocity diagram (minus sign for tensile stress). d, Free surface velocity history.

Finally, the authors conflate the spall strength - which is a mechanistic quantity, not a material property - with the ideal strength. In most metals, the spall strength is a multiple of the yield strength, sometimes 3-5 times the yield strength. The purported unusual behavior here is not sensible in this light.

- *The referee is correct that for most metals (monocrystalline and polycrystalline), the fracture strength (sometimes known as the ultimate strength or spall strength depending on the experimental conditions) is usually higher than the yield strength. However, we respectfully disagree with the referee's argument that the spall strength should not be used to evaluate the ideal strength. Ideal strength of a material refers to the maximum possible stress that the material can withstand before failure (either under quasi-static or dynamic conditions). In this sense, even though the specimen is not in a uniform stress state throughout the thickness in plate impact experiments or laser induced shock experiments, the tensile stress that causes the material to spall ought to be considered as the fracture strength and can be used to assess the ideal strength of the material. It is worth noting that both the tensile strength and the shear strength (yield strength) have their theoretical upper limits, as brought up by Orowan [18] and Frenkel [19], respectively.*
- *In fact, the spall strengths of monocrystalline and polycrystalline metals are higher than the yield strengths precisely because plastic deformation mechanisms (e.g., dislocations and twinning) will be activated far earlier than failure even at the extreme strain rates similar to those used in our study. In other words, monocrystalline and polycrystalline metals would yield first and then fail. In contrast, $\text{Cu}_{50}\text{Zr}_{50}$ tested at $1.4\text{--}2.8 \times 10^7 \text{ s}^{-1}$ did not show signs of the classical plasticity mechanisms, phase transformation, or shear banding before failure. The predominant failure mechanism was the material instability caused by cavitation, as evidenced by the XRD, SEM, and TEM characterizations, and supported by molecular dynamics simulations.*
- *Therefore, we believe that the ultrahigh spall/fracture stress (approaching $\sim E/6$ and $\sim M/13$) and the unique failure mechanism for $\text{Cu}_{50}\text{Zr}_{50}$ MG at the extreme condition discovered in this study are innovative and impactful to the field of mechanics of materials.*

References

- [1] M.-B. Tang, D.-Q. Zhao, M.-X. Pan, W.-H. Wang, Binary Cu-Zr Bulk Metallic Glasses, Chinese Physics Letters 21(5) (2004) 901.
- [2] F. Zhu, S. Song, K.M. Reddy, A. Hirata, M. Chen, Spatial heterogeneity as the structure feature for structure-property relationship of metallic glasses, Nat Commun 9(1) (2018) 3965.
- [3] D. Xu, G. Duan, W.L. Johnson, Unusual Glass-Forming Ability of Bulk Amorphous Alloys Based on Ordinary Metal Copper, Physical Review Letters 92(24) (2004) 245504.
- [4] G. Ding, C. Li, A. Zaccone, W.H. Wang, H.C. Lei, F. Jiang, Z. Ling, M.Q. Jiang, Ultrafast extreme rejuvenation of metallic glasses by shock compression, Science Advances 5(8) (2019).
- [5] S.N. Luo, B.J. Jensen, D.E. Hooks, K. Fezzaa, K.J. Ramos, J.D. Yeager, K. Kwiatkowski, T. Shimada, Gas gun shock experiments with single-pulse x-ray phase contrast imaging and diffraction at the Advanced Photon Source, Rev Sci Instrum 83(7) (2012) 073903.
- [6] D.R. Jones, D.J. Chapman, D.E. Eakins, A gas gun based technique for studying the role of temperature

in dynamic fracture and fragmentation, *Journal of Applied Physics* 114(17) (2013).

[7] J. Coakley, A. Higginbotham, D. McGonegle, J. Wark, Femtosecond quantification of void evolution during rapid material failure, *Science Advances* 6 (2020) eabb4434.

[8] P. Wen, B. Demaske, D.E. Spearot, S.R. Phillpot, Shock compression of $\text{Cu}_x\text{Zr}_{100-x}$ metallic glasses from molecular dynamics simulations, *Journal of Materials Science* 53(8) (2017) 5719-5732.

[9] W.L. Johnson, K. Samwer, A universal criterion for plastic yielding of metallic glasses with a $(T/T_g)^{2/3}$ temperature dependence, *Phys Rev Lett* 95(19) (2005) 195501.

[10] X.C. Tang, C. Li, H.Y. Li, X.H. Xiao, L. Lu, X.H. Yao, S.N. Luo, Cup-cone structure in spallation of bulk metallic glasses, *Acta Materialia* 178 (2019) 219-227.

[11] H. Ehsani, J.D. Boyd, J. Wang, M.E. Grady, Evolution of the laser-induced spallation technique in film adhesion measurement, *Applied Mechanics Reviews* 73(3) (2021).

[12] T. Antoun, L. Seaman, D.R. Curran, G.I. Kanel, S.V. Razorenov, A.V. Utkin, *Spall fracture*, Springer Science & Business Media 2003.

[13] S. Novikov, I. Divnov, A. Ivanov, The study of fracture of steel, aluminium and copper under explosive loading, *Fizika metallov i Metallovedeniye* 21(4) (1966) 608-615.

[14] E. Dekel, S. Eliezer, Z. Henis, E. Moshe, A. Ludmirsky, I. Goldberg, Spallation model for the high strain rates range, *Journal of applied physics* 84(9) (1998) 4851-4858.

[15] H. Jarmakani, B. Maddox, C.T. Wei, D. Kalantar, M.A. Meyers, Laser shock-induced spalling and fragmentation in vanadium, *Acta Materialia* 58(14) (2010) 4604-4628.

[16] T.P. Remington, E.N. Hahn, S. Zhao, R. Flanagan, J.C.E. Mertens, S. Sabbaghianrad, T.G. Langdon, C.E. Wehrenberg, B.R. Maddox, D.C. Swift, B.A. Remington, N. Chawla, M.A. Meyers, Spall strength dependence on grain size and strain rate in tantalum, *Acta Materialia* 158 (2018) 313-329.

[17] G. Righi, C.J. Ruestes, C.V. Stan, S.J. Ali, R.E. Rudd, M. Kawasaki, H.-S. Park, M.A. Meyers, Towards the ultimate strength of iron: spalling through laser shock, *Acta Materialia* (2021).

[18] E. Orowan, Fracture and strength of solids, *Reports on Progress in Physics* 12(1) (1949) 185.

[19] J. Frenkel, Zur theorie der elastizitätsgrenze und der festigkeit kristallinischer körper, *Zeitschrift für Physik* 37(7-8) (1926) 572-609.

REVIEWER COMMENTS

Reviewer #1 (Remarks to the Author):

The reviewer finds the work presented by the authors very interesting both in the material studied and the testing technique utilized to probe its material response and properties. Furthermore, the authors have done a very thorough job of addressing the reviewers' initial questions and concerns specifically regarding additional microstructural characterization supporting their conclusion of the failure mechanism being void nucleation and growth controlled. The reviewer believes the manuscript is acceptable for publication but would advise the authors to address a few minor questions and concerns listed below:

In the 3rd paragraph of the results section, the authors list Figure 1 when discussing micro-voids in the specimen shocked at 5.0 J laser pulse. This should be in reference to the SEM micrographs in Figure 3a.

When viewing the SEM micrograph in Figure 3a, there is a discolored band above and to the left of the coalesced void. Can the authors comment on this feature? Did they happen to do chemical analysis of this band feature? Also did the authors consider doing ion contrast imaging using the Ga ion source on the sidewalls of the exposed surfaces?

The authors note the critical void diameter to be approximately 7.3 nm from their MD simulations. The reviewer would suspect this to be smaller than the shock width generated during their laser shock experiments. Can the authors estimate/confirm the shock width generated during their experiments?

Also, can the authors comment on the amount of strain imparted on their specimen using their laser shock setup?

Finally, the authors mention in a few places within the manuscript that plastic deformation being mediated by dislocations for both single-crystal and nanocrystalline metals. While this is true for single-crystal and coarse grained, polycrystalline materials, nanocrystalline materials deformation mechanism are generally dominated by grain boundary mechanisms such as sliding and rotation due to the high-volume fraction of the material they compose in nanocrystalline metals. As a researcher studying nanocrystalline metals, it is worth clarifying these statements within the manuscript in my opinion.

Reviewer #4 (Remarks to the Author):

Comments on the author's response to the first reviewer:

1. We doubt that cavitation is the primary failure mechanism for amorphous alloys under laser loading. According to the relevant experimental data [1], the fracture surface morphology is indeed rougher at extremely high strain rates ($>10^7$ /s) as compared to the outcomes of gas gun studies (105–106 /s). Nonetheless, shear bands and voids continue to coexist, which contradicts the conclusion reached in this study. We believe there may be the following reasons :
Firstly, compared to 800 J in [1], the laser energy used in this article is relatively low (2-15 J). It is widely known [2] that the impact velocity determines the fracture morphology of amorphous alloys, which can range from voids to shear bands. Consequently, the results of this study may represent only exceptional cases within a particular energy range.
Secondly, this article examines only one composition and component of amorphous alloys. The conclusions may not apply to all amorphous alloy systems. While summarizing, greater care should be taken. We would recommend that the author provide more systematized experimental and simulation results, or at the very least revise the paper's inappropriate descriptions.

2. Regarding why the rate sensitivity of MD simulation is lower than the experimental law, the author neglected to mention the crucial influence of the potential function. Concerning the problems caused by high cooling rates, on the one hand, the author adopted a cooling rate that is two orders of magnitude greater than the current level of affordable computing resources [3]. Alternatively, if the author wishes to test whether a lower cooling rate can produce a more experimental pattern, there are technical means [4] that can produce the same effect as an exceedingly low cooling rate.

3. At this strain rate, we believe it is possible for the amorphous alloy suspected by the reviewer to manifest extremely high spall strength (on the order of 10 GPa). On the one hand, as the author points out, there is already available experimental evidence in crystals. The study results we already know about [5] also back up this rule, as shown below.

On the other hand, the Exponential type of strength growth is brought about by the viscosity (rate effect) of material plastic deformation, the inertia effect of damage evolution, and the relativistic effect [6]. The existence of these mechanisms is independent of the material, and they are universal for both crystalline and amorphous substances, with only differences in their specific forms. Therefore, we are not shocked by the work's conclusion based on this knowledge. A sufficiently high strength can result in a modulus-to-strength ratio that exceeds the theoretical limit, given that the modulus of amorphous systems is typically lower than that of crystals of the same composition.

Comments on the author's response to the second reviewer:

We believe that the author's choice of 0.4 for the Poisson's ratio of the Cu50Zr50 system is still questionable, and that the author's inspection work is not systematic. On the one hand, the author's experimental work [7] did indeed yield a measurement value of 0.4. However, this does not imply that the samples used by the author are identical, as preparation processes and techniques vary. On the other hand, it is well known that the results of an MD simulation differ from those of an experiment and that the cooling rate has a significant effect on these differences. In addition, there are numerous methods for calculating Poisson's ratio using MD, and the results obtained vary.

We recommend that the authors directly measure the longitudinal wave sound velocity C_l and transverse wave sound velocity C_t of materials using ultrasonic instruments, and then calculate the Poisson's ratio $\nu = 1/2(1 - C_l^2/C_t^2)$ and modulus $E = 2\rho C_t^2(1 + \nu)$ of materials based on these measurements, which will more accurately reflect the actual situation of the materials they use.

Comments on the author's response to the third reviewer:

Concerning whether stable shock waves can form in such a brief period of time as the reviewer suspects, MD simulation can play a role in testing, for example by outputting a profile of the stress wave propagation process. However, we believe that the absence of shock wave formation has no impact on the acquisition of strain rate information from free surface velocity data. Although the accuracy is questionable, the higher strain rate compared to gas gun loading should not be an issue. As for the issue of whether the estimation of strain rate is reasonable, it is a fairly challenging question to determine. In addition to free surface velocity, it is challenging to acquire other information for evaluating strain rate and spalling strength using current experimental techniques. We acknowledge that the method used by the author is indeed a universal method. However, this approach has severe limitations. Technically speaking, it analyzes strain rate during compression unloading rather than tensile loading, but the difference between the two is mercifully not too great. However, the situation is significantly worse in terms of spalling strength. The spalling intensity reflected by the free surface velocity is not identical to the local stress amplitude that actually causes sample fracture, and is influenced by a number of variables, besides strain rate.

On the other hand, due to the continuously changing slope during the propagation of sparse waves, the strain rate and spalling strength measured from the free surface velocity are in fact proportional to the sample's thickness. In cases where the sample thickness is fixed, it is possible to disregard the thickness factor and compare data from the same researcher without regard to the factor. However, because the data compared in this article encompasses several experimental procedures and is done by different researchers, the thickness correction hypothesis must be used. This argument was also

made in the fourth section of the sources the author [8] mentioned.

Concerning the rate effect and stress amplitude coupling issues raised by reviewers, we believe there is some validity. The spalling strength is influenced primarily by strain rate, but to a lesser extent also by stress amplitude. There is relevant research [9] on this topic, but it has not yet been examined in all materials. It is suggested that the author exercise greater caution when discussing.

References:

- [1] Jodar B, Loison D, Yokoyama Y, et al. Localized atomic segregation in the spalled area of a Zr₅₀Cu₄₀Al₁₀ bulk metallic glasses induced by laser-shock experiment[J]. Journal of Physics D: Applied Physics, 2018, 51(6): 065304.
- [2] Escobedo J P, Gupta Y M. Dynamic tensile response of Zr-based bulk amorphous alloys: Fracture morphologies and mechanisms[J]. Journal of Applied Physics, 2010, 107(12).
- [3] Guan P, Lu S, Spector M J B, et al. Cavitation in amorphous solids[J]. Physical Review Letters, 2013, 110(18): 185502.
- [4] Ninarello A, Berthier L, Coslovich D. Models and algorithms for the next generation of glass transition studies[J]. Physical Review X, 2017, 7(2): 021039.
- [5] Wilkerson J W. On the micromechanics of void dynamics at extreme rates[J]. International Journal of Plasticity, 2017, 95: 21-42.
- [6] Wilkerson J W, Ramesh K T. A dynamic void growth model governed by dislocation kinetics[J]. Journal of the Mechanics and Physics of Solids, 2014, 70: 262-280.
- [7] Johnson W L, Samwer K. A universal criterion for plastic yielding of metallic glasses with a $(T/T_g)^{2/3}$ temperature dependence[J]. Physical review letters, 2005, 95(19): 195501.
- [8] Antoun T. Spall fracture[M]. Springer Science & Business Media, 2003.
- [9] Li C, Li B, Huang J Y, et al. Spall damage of a mild carbon steel: Effects of peak stress, strain rate and pulse duration[J]. Materials Science and Engineering: A, 2016, 660: 139-147.

Response to the editors and reviewers

We thank the editor and reviewers very much for the feedback and constructive comments. Changes have been made in the revised manuscript (text highlighted in yellow) based on reviewers' comments. All page numbers in our responses refer to the revised manuscript unless otherwise stated. The following are the itemized responses to the reviewers' comments.

Comments from the editors and reviewers:

Reviewer #1

The reviewer finds the work presented by the authors very interesting both in the material studied and the testing technique utilized to probe its material response and properties. Furthermore, the authors have done a very thorough job of addressing the reviewers' initial questions and concerns specifically regarding additional microstructural characterization supporting their conclusion of the failure mechanism being void nucleation and growth controlled. The reviewer believes the manuscript is acceptable for publication but would advise the authors to address a few minor questions and concerns listed below.

- *We thank the referee for the recommendation.*

In the 3rd paragraph of the results section, the authors list Figure 1 when discussing micro-voids in the specimen shocked at 5.0 J laser pulse. This should be in reference to the SEM micrographs in Figure 3a.

- *We thank the referee for pointing out the mistake. We have double-checked and corrected the figure references.*

When viewing the SEM micrograph in Figure 3a, there is a discolored band above and to the left of the coalesced void. Can the authors comment on this feature? Did they happen to do chemical analysis of this band feature? Also did the authors consider doing ion contrast imaging using the Ga ion source on the sidewalls of the exposed surfaces?

- *We thank the referee for the instructive suggestion. We performed energy dispersive X-ray spectroscopy (EDS) elemental mapping and ion contrast imaging using Ga ion source on the sidewalls around the nanovoids shown in Fig 3b. The EDS map suggests no segregation and aggregation of elements near the nanovoids. The discolored bands near the coalesced void were much less obvious under the ion contrast imaging mode. Therefore, we suspect that the contrast differences in Figure 3 were from the uneven material thickness along the direction perpendicular to the sidewall as the result of the coalescence of multiple three-dimensional voids.*
- *We have added the discussion on Page 3 and Supplementary Fig. 8 as shown below:*

on Page 3:

Further, energy dispersive X-ray spectroscopy (EDS) elemental mapping and ion contrast imaging using Ga ion source on the sidewalls showed no segregation and aggregation of elements near the nanovoids (Supplementary Fig. 8a and b). The band feature depicted in the magnified image in **Error! Reference source not found.** correlates to cavity calescence, which were identified more clearly by ion contrast imaging (Supplementary Fig. 8c).

Supplementary Fig. 8. Chemical analysis and ion contrast imaging of the void structures. a and b, SEM image and the energy dispersive X-ray spectroscopy (EDS) elemental map of the nanovoid region in Figure 3b in the main text. c, Ion contrast images of the sidewall and nanovoids.

The authors note the critical void diameter to be approximately 7.3 nm from their MD simulations. The reviewer would suspect this to be smaller than the shock width generated during their laser shock experiments. Can the authors estimate/confirm the shock width generated during their experiments?

- *The referee was correct that the critical void diameter is significantly smaller than the shock width. It reflects the local nature of damage and cavitation. We can evaluate the shock width d based on the shock duration τ extracted from FSV curves (approximately equal to the time span from the beginning of the shock pulse till the pullback signal). Given $\tau \approx 3-5$ ns, the shock width is estimated as $d = c\tau \approx 15-20$ μm.*

- *We have added the following discussion about the shock width on Page 7:*

The estimated shock width ranged from 15 to 20 μm (determined based on the speed of sound and the duration of the shock that lasted approximately 3 to 5 ns). The shock width is significantly larger than the local critical void diameter, D_c , by several orders of magnitude.

Also, can the authors comment on the amount of strain imparted on their specimen using their laser shock setup?

- *We thank the referee for the suggestion. The strain at the spallation was estimated as $\epsilon \approx \Delta v_{fs} / (2c)$. For the laser energies used in our study, the corresponding strain ϵ ranges from 4.5% to 7.7%.*
- *We have added the following discussion about the strain on Page 7:*

The tensile strains imparted on the MG samples were estimated as _____, ranging from 4.5% to 7.7%.

Finally, the authors mention in a few places within the manuscript that plastic deformation being mediated by dislocations for both single-crystal and nanocrystalline metals. While this is true for single-crystal and coarse grained, polycrystalline materials, nanocrystalline materials deformation mechanism are generally dominated by grain boundary mechanisms such as sliding and rotation due to the high-volume fraction of the material they compose in nanocrystalline metals. As a researcher studying nanocrystalline metals, it is worth clarifying these statements within the manuscript in my opinion.

- *We thank the referee for the suggestion. We have revised the discussion in the 2nd paragraph on Page 2 to clarify this point as follows:*

This is because single-crystal **or** nanocrystalline metals undergo extensive plastic deformation mediated by dislocation movements, **grain rotation, and grain boundary sliding** (predominant at low or moderate strain rates^{18, 19}) and mechanical twinning (predominant at ultrahigh strain rates¹⁷) prior to failure, which accounts for the notable difference between experimental and ideal strength values.

Reviewer #4

Comments on the author's response to the first reviewer:

1. We doubt that cavitation is the primary failure mechanism for amorphous alloys under laser loading. According to the relevant experimental data [R1], the fracture surface morphology is indeed rougher at

extremely high strain rates ($>10^7$ /s) as compared to the outcomes of gas gun studies (10^5 – 10^6 /s). Nonetheless, shear bands and voids continue to coexist, which contradicts the conclusion reached in this study. We believe there may be the following reasons:

Firstly, compared to 800 J in [R1], the laser energy used in this article is relatively low (2-15 J). It is widely known [R2] that the impact velocity determines the fracture morphology of amorphous alloys, which can range from voids to shear bands. Consequently, the results of this study may represent only exceptional cases within a particular energy range.

- *We thank the reviewer for bringing up this point. We carefully read the relevant experimental studies in Ref. [R1]. We believe that the finding in Ref. [R1] does not contradict with our results. The reasons are following.*
- *The strain rate was not characterized in Ref. [R1]. The SEM images of the fracture surface in Ref. [R1] actually showed void dominated features, which is comparable to our study. However, the average void diameter in Ref. [R1] was much larger, $\sim 10\ \mu\text{m}$, compared to the range of $0.5\ \mu\text{m}$ – $3\ \mu\text{m}$ in our study. There was no strong evidence of shear banding in Ref. [R1], and the fracture morphology was very different from the cup-cone structure discovered in Ref. [R5] which was the sign of the coexistence of shear-banding and void growth (see Fig. R1). Thus, it is possible that the strain rates in Ref. [R1] might have exceeded the transition strain rate at which void growth mechanism predominates, but were still lower than those in our tests.*
- *Our hypothesis was drawn from the fact that the tensile strain rate is more sensitive to the laser duration than the laser energy – a higher laser duration would lead to a slower strain rate [R3]. The laser pulse duration in Ref. [R1] was 4.6 ns, which was 2-4 times longer than the laser durations in our study. Moreover, the sample thickness in Ref. [R1] was $215\ \mu\text{m}$, roughly double the thickness of our samples. Since the laser-induced pressure pulse decays during propagation to the back free surface, a thicker sample would experience a lower strain rate [R4]. Therefore, we think that the strain rate in Ref. [R1], although much greater than those in Ref. [R5], was still lower than those in our tests.*

Fig. R1 Spall surface morphologies adopted from Jodar et al. [R1] (a) versus that from Tang et al. [R5] (b). In (b), both cup and cone were observed on the same spall plane.

Secondly, this article examines only one composition and component of amorphous alloys. The conclusions may not apply to all amorphous alloy systems. While summarizing, greater care should be taken. We would recommend that the author provide more systematized experimental and simulation

results, or at the very least revise the paper's inappropriate descriptions.

- *We thank the referee for this suggestion. We agree with the referee that the universality of our finding still needs to be verified by further studies. We have revised the discussions in the conclusion on Page 6 as follows:*

Future research should aim to explore the mechanical properties of other amorphous alloys at extreme strain rates to demonstrate the potential universality of the ultrahigh strength observed on Cu-Zr MGs herein.

2. Regarding why the rate sensitivity of MD simulation is lower than the experimental law, the author neglected to mention the crucial influence of the potential function. Concerning the problems caused by high cooling rates, on the one hand, the author adopted a cooling rate that is two orders of magnitude greater than the current level of affordable computing resources [R6]. Alternatively, if the author wishes to test whether a lower cooling rate can produce a more experimental pattern, there are technical means [R7] that can produce the same effect as an exceedingly low cooling rate.

- *We thank the reviewer for bringing up these important points. To check the influences of interatomic potential function and the cooling rate on the results, we carried out more MD simulations.*
- *In the last manuscript, we prepared the large $\text{Cu}_{50}\text{Zr}_{50}$ MG model (containing 32 million atoms) using a high cooling rate of $\sim 10^{12}$ K/s. For such a large system, it is computationally very challenging to reduce the cooling rate down to the same level as in [R6]. Instead, we followed the same approach in a recent study [R8] to prepare three smaller ($20 \times 20 \times 20 \text{ nm}^3$) $\text{Cu}_{50}\text{Zr}_{50}$ models (containing 0.5 million atoms) at three cooling rates of 1.7×10^{12} , 6.8×10^{11} and 1.7×10^{11} K/s through direct MD simulations. We then used a logarithmic function to fit the relation between the energy per atom at 300 K and the cooling rate. Next, we performed hybrid MD/Monte Carlo (MC) simulations in the variance constrained semi-grand canonical ensemble implemented in LAMMPS. The energies per atom at 300K obtained from the hybrid MD/MC simulations were used to determine the corresponding effective cooling rates by extrapolating the logarithmic relationship obtained from the direct MD simulations, as shown in Supplementary Fig. 12a. This way, we obtained four $\text{Cu}_{50}\text{Zr}_{50}$ MG models ($20 \times 20 \times 20 \text{ nm}^3$) at much lower effective cooling rates (2.1×10^{11} K/s, 2.0×10^{10} K/s, 8.6×10^9 K/s and 1.1×10^7 K/s). Finally, we performed uniaxial strain tensile tests on the four MG models at two strain rates ($5.0 \times 10^8 \text{ s}^{-1}$ and $5.0 \times 10^9 \text{ s}^{-1}$) using the embedded-atom method (EAM) interatomic potential developed by Mendeleev et al. The results give almost the same strain rate sensitivity as that of the large model ($20 \times 20 \times 20 \text{ nm}^3$) which was cooled at 1.7×10^{12} K/s (Supplementary Fig. 12b).*
- *On the other hand, to check how different potential functions affect our results, we adopted two EAM potential functions [R9, R10] and a MEAM potential function [R11] to repeat the uniaxial strain tensile tests on the $20 \times 20 \times 20 \text{ nm}^3$ $\text{Cu}_{50}\text{Zr}_{50}$ model at two strain rates ($5.0 \times 10^8 \text{ s}^{-1}$ and $5.0 \times 10^9 \text{ s}^{-1}$). As shown in Supplementary Fig. 12c and d, although different potential functions gave different elastic moduli and material strengths, the strain rate sensitivity is not sensitive to the choice of potential functions.*

- Thus, the additional MD simulations suggest that the cooling rate used to generate the model and the choice of potential functions are not the dominating factors for the lower strain rate sensitivity. In contrast, introducing a small defect into the model effectively brings up the strain sensitivity, as shown in Fig. 1b. Therefore, we attribute the high strain rate sensitivity observed in the experiments to the inherent defects in MG.
- We have added the following discussions on Page 4 and the Methods section on Page 8, and added the Supplementary Fig. S12:
On Page 4:

We carried out additional computational studies of the effects of cooling rate and potential function, both of which exhibited minor influences on the strain rate sensitivity (Supplementary Fig. 12). These results show that the spall strength and the strain rate sensitivity of MGs is highly sensitive to their internal structural defects.

On Page 8:

To check the effect of the cooling rate on the strain rate sensitivity, three smaller ($20 \times 20 \times 20 \text{ nm}^3$) $\text{Cu}_{50}\text{Zr}_{50}$ MG models at different cooling rates (1.7×10^{12} , 6.8×10^{11} and $1.7 \times 10^{11} \text{ K/s}$) were prepared (Supplementary Fig. 12a). We then used a logarithmic function to fit the relation between the energy per atom at 300 K and the cooling rate. Next, we performed hybrid MD/Monte Carlo (MC) simulations in the variance constrained semi-grand canonical ensemble, following the method in ref. 41. The energies per atom at 300K obtained from the hybrid MD/MC simulations were used to determine the corresponding effective cooling rates by extrapolating the logarithmic relationship obtained from the direct MD simulations. This way, we obtained four $\text{Cu}_{50}\text{Zr}_{50}$ MG models ($20 \times 20 \times 20 \text{ nm}^3$) at much lower effective cooling rates ($2.1 \times 10^{11} \text{ K/s}$, $2.0 \times 10^{10} \text{ K/s}$, $8.6 \times 10^9 \text{ K/s}$ and $1.1 \times 10^7 \text{ K/s}$). Finally, we performed uniaxial strain tensile tests on the four MG models at two strain rates ($5.0 \times 10^8 \text{ s}^{-1}$ and $5.0 \times 10^9 \text{ s}^{-1}$) using the embedded-atom method (EAM) interatomic potential developed by Mendeleev *et al.*⁶¹. The results show a strain rate sensitivity that is nearly identical to that of the larger model ($80 \times 80 \times 80 \text{ nm}^3$) which was cooled at $1.7 \times 10^{12} \text{ K/s}$ (Supplementary Fig. 12b).

To whether different interatomic potential functions would affect the strain rate sensitivity, we performed additional uniaxial strain tensile tests on the smaller $\text{Cu}_{50}\text{Zr}_{50}$ model ($20 \times 20 \times 20 \text{ nm}^3$, $1.7 \times 10^{12} \text{ K/s}$) using two EAM potential functions^{61, 63} and a modified embedded-atom method (MEAM) potential function⁶⁴. Two tensile strain rates ($5 \times 10^8 \text{ s}^{-1}$ and $5 \times 10^9 \text{ s}^{-1}$) were applied. Supplementary Fig. 12c and d demonstrate that several potential functions had similar strain rate sensitivities, despite variations in elastic moduli and material strengths.

Supplementary Fig. 12 Effects of interatomic potential and cooling rate in MD simulations. a, The per atom energy at 300 K vs. effective cooling rate for the Cu-Zr sample containing ~ 0.5 million atoms. **b,** Summary of the data for strength vs. tensile strain rate showing the cooling rate barely changing the strain rate sensitivity. The data from Fig.1 (black circles) was extracted for comparison. **c,** Stress-strain curves of Cu-Zr samples using different interatomic potentials deformed at the strain rate of 5.0×10^8 s⁻¹ (solid lines) and 5.0×10^9 s⁻¹ (dashed lines). **d,** Summary of the data for strength vs. tensile strain rate showing the choice of interatomic potential barely changing the strain rate sensitivity. The data from Fig.1 (black circles) was extracted for comparison.

3. At this strain rate, we believe it is possible for the amorphous alloy suspected by the reviewer to manifest extremely high spall strength (on the order of 10 GPa). On the one hand, as the author points out, there is already available experimental evidence in crystals. The study results we already know about [R12] also back up this rule, as shown below.

On the other hand, the Exponential type of strength growth is brought about by the viscosity (rate effect) of material plastic deformation, the inertia effect of damage evolution, and the relativistic effect [R13]. The existence of these mechanisms is independent of the material, and they are universal for both crystalline and amorphous substances, with only differences in their specific forms. Therefore, we are not shocked by the work's conclusion based on this knowledge. A sufficiently high strength can result in a modulus-to-strength ratio that exceeds the theoretical limit, given that the modulus of amorphous

systems is typically lower than that of crystals of the same composition.

- *We agree that the phenomena of increased strength with higher strain rates is a universally observed trend. Nevertheless, there exist differences in the mechanics underlying the enhancement of strength between crystalline and amorphous materials. In the following discussions, we want to provide further elucidation on the influence of inertia, relativity, and viscosity, as highlighted by the referee.*
- *First, the inertia effect on the damage evolution is related to the transient void growth as described by the Rayleigh-Plesset theory: $\frac{da}{dt} = \frac{2}{3} \frac{R}{\rho} \left(\frac{p_\infty - p}{R} \right)^{3/2}$, where a is the void radius, ρ is the density, p_∞ is the far-field hydrostatic stress and R is the void growth resistance [R12]. When the void radius a is much smaller than the transition void size a_{trans} , the inertia effect can be neglected and the above equation reduces to $\frac{da}{dt} = \frac{2}{3} \frac{R}{\rho} \left(\frac{p_\infty - p}{R} \right)^{3/2}$. Wilkerson and Ramesh [R13] showed that a_{trans} is approximately 100 nm, which is significantly greater than the critical void size at material failure (~ 10 nm). Hence, the inertia effect has been ignored in the present study, similar to the majority of void growth models employed in spallation studies [R14-R16].*
- *Second, it should be noted that the relativistic effect imposes a constraint on the dislocation velocity, v , which is limited by the shear wave speed, c_s . The dislocation drag coefficient is expressed as $B = B_0 \left(\frac{v}{c_s} \right)^2$ (B is the drag coefficient at the low speed limit). At extremely high strain rates, the dislocation velocity (v) attains the upper limit (c_s) and does not exhibit further augmentation as the loading increases. Consequently, this results in a relatively gradual expansion of voids generated by dislocations, leading to a corresponding rise in strength (note that that the relativistic effect for crystalline metals has been taken into account in Eq. (3) in the main text). Nevertheless, it is expected that amorphous alloys would not show this characteristic due to the absence of preferred slip directions (also referred as slip planes in crystalline metals).*
- *Lastly, the phenomenon of viscosity effect, also known as viscoplasticity, contributes to the enhancement of strength in both crystalline and amorphous metals, but with distinct atomistic mechanisms. The control of viscoplasticity in a crystal is governed by the drag coefficient B , which influences the motion of dislocations and subsequently affects the formation of voids. On the other hand, the influence of viscosity on the void growth in an amorphous alloy is regulated by the mobility coefficient, m . This coefficient represents the rate at which atomic rearrangement occurs collectively to form shear or tension transformation zones.*
- *Overall, the primary focus of our study is to highlight the distinctive void growth kinetics (as presented by Eq. (1) and Eq. (3) in the main text, respectively) in differentiating the strength characteristics of an amorphous alloy from those of a crystalline metal. Our analysis demonstrates that the void growth rate of the Cu-Zr amorphous alloy is nearly two orders of magnitude slower than that of crystalline metals. Thus, Cu-Zr amorphous alloy exhibits substantially improved spall strength, as explained in detail on Page 6. In this regard, the present study offers novel perspectives on the failure mechanism in amorphous alloys under extreme strain rates, in contrast to that*

observed in crystalline metals. We believe that these findings hold considerable significance for the fields of mechanics and materials science.

Comments on the author's response to the second reviewer:

We believe that the author's choice of 0.4 for the Poisson's ratio of the Cu₅₀Zr₅₀ system is still questionable, and that the author's inspection work is not systematic. On the one hand, the author's experimental work [R17] did indeed yield a measurement value of 0.4. However, this does not imply that the samples used by the author are identical, as preparation processes and techniques vary. On the other hand, it is well known that the results of an MD simulation differ from those of an experiment and that the cooling rate has a significant effect on these differences. In addition, there are numerous methods for calculating Poisson's ratio using MD, and the results obtained vary.

We recommend that the authors directly measure the longitudinal wave sound velocity C_l and transverse wave sound velocity C_t of materials using ultrasonic instruments, and then calculate the Poisson's ratio $\nu = 1/2(1 - C_l^2/C_t^2)$ and modulus $E = 2\rho C_t^2(1 + \nu)$ of materials based on these measurements, which will more accurately reflect the actual situation of the materials they use.

- We appreciate the referee's insightful suggestion. We did not use the ultrasound method due to the very small sample thickness. Instead, we directly measured the elastic modulus of our Cu₅₀Zr₅₀ MG ribbon using in-situ SEM tensile tests. The strain was measured using the digital image correlation (DIC) method. From the stress-strain curves, we obtained the elastic modulus of Cu₅₀Zr₅₀ MG, $E = 65.9 \pm 2.6$ GPa. Using the newly measured elastic modulus and the previously determined P-wave modulus $M = 149.5$ GPa, the Poisson's ratio of our Cu₅₀Zr₅₀ MG is determined to be $\nu = 0.407$, very close to our previous estimation from the MD results.*
- We have revised Fig. 4 and the text on Page 3. We also added the Supplementary Fig. 16 and the details in the Methods section on Page 7:
On Page 3:*

Note that the Young's modulus (measured by uniaxial tensile test; the corresponding Poisson's ratio, agreeing with the literature²⁹), thus, the material strength approached approximately $E/6$.

On Page 7:

Tensile Tests

Uniaxial tensile tests at room temperature were conducted on a micro-tester (MT300, Deben) with a nominal strain rate of $1.5 \times 10^{-4} \text{ s}^{-1}$ under SEM (Phenom XL G2, Thermo Fisher Scientific). The Cu₅₀Zr₅₀ ribbon sample with a cross-section of $2.3 \times 0.053 \text{ mm}^2$ was used. The tensile strain was obtained using the open-source digital image correlation MATLAB code, Ncorr⁵⁹. The Young's modulus was determined to be 65.9 ± 2.6 GPa from the stress-strain curves (Supplementary Fig. 16).

Fig. 4 Summary of the measured ultimate strengths for metallic materials. Representative experimental results for crystalline and amorphous metals tested under quasi-static conditions^{8, 9, 10, 23, 49, 50, 51}, at high rates ($10^4 \text{ s}^{-1} < < 10^6 \text{ s}^{-1}$)^{25, 26, 27, 28}, and ultrahigh rates ($> 10^6 \text{ s}^{-1}$)^{11, 14, 15, 16, 17, 52} are included for comparison. Our study on Cu-Zr MG shows the record-setting spall strength of $\sim M/13$ for metals.

Supplementary Fig. 16. Stress-strain curves of the $\text{Cu}_{50}\text{Zr}_{50}$ MG ribbon subjected to quasi-static uniaxial tensile tests.

Comments on the author's response to the third reviewer:

Concerning whether stable shock waves can form in such a brief period of time as the reviewer suspects, MD simulation can play a role in testing, for example by outputting a profile of the stress wave propagation process. However, we believe that the absence of shock wave formation has no impact on the acquisition of strain rate information from free surface velocity data. Although the accuracy is questionable, the higher strain rate compared to gas gun loading should not be an issue.

As for the issue of whether the estimation of strain rate is reasonable, it is a fairly challenging question to determine. In addition to free surface velocity, it is challenging to acquire other information for evaluating strain rate and spalling strength using current experimental techniques. We acknowledge that the method used by the author is indeed a universal method. However, this approach has severe limitations. Technically speaking, it analyzes strain rate during compression unloading rather than tensile loading, but the difference between the two is mercifully not too great. However, the situation is significantly worse in terms of spalling strength. The spalling intensity reflected by the free surface velocity is not identical to the local stress amplitude that actually causes sample fracture, and is influenced by a number of variables, besides strain rate.

On the other hand, due to the continuously changing slope during the propagation of sparse waves, the strain rate and spalling strength measured from the free surface velocity are in fact proportional to the sample's thickness. In cases where the sample thickness is fixed, it is possible to disregard the thickness factor and compare data from the same researcher without regard to the factor. However, because the data compared in this article encompasses several experimental procedures and is done by different researchers, the thickness correction hypothesis must be used. This argument was also made in the fourth section of the sources the author [R18] mentioned.

Concerning the rate effect and stress amplitude coupling issues raised by reviewers, we believe there is some validity. The spalling strength is influenced primarily by strain rate, but to a lesser extent also by stress amplitude. There is relevant research [R19] on this topic, but it has not yet been examined in all materials. It is suggested that the author exercise greater caution when discussing.

- *We would like to respond to the above comments together. First, stable shock waves cannot form in laser shock experiments, as discussed in our last response to the reviewers. However, it does not influence the evaluation of the local strength of materials.*
- *We agree with the referee that although the strain rate is the dominant factor for the material strength, the thickness correction is necessary since the experiments mentioned in this study were done by different groups.*
- *Following the referee's suggestion, we used Stepanov's thickness correction formula*

(C_{sp} is the correction factor, v_{fs} is the free surface

velocity derivative ahead of the spall pulse, and h_{sp} is the thickness of the spalling layer) [R20]. This formula is suitable for the case of triangular impulse [R18, R21]. Based on the cross-sectional SEM image (Fig. 3a), we estimated the thickness of the spalling layer $h_{sp} \approx 10 \mu\text{m}$. As listed in Supplementary Table 1 in the revision, the thickness correction only introduced a minor change in the strength values (the maximum relative variation $\approx 6.2\%$). Thus, we believe the linear acoustic approximation adopted here is still valid. We have added more discussions in the Methods section on Page 7 as follows:

We also employed the formula with the sample thickness correction to calculate the spall strengths^{57, 58}. The thickness correction formula gave a slightly higher spall strengths, but the difference between two methods is less than 6.2% (see Supplementary Table 1). Therefore, we still report the results using the classic formula $\sigma_s = \rho c \Delta v_{fs} / 2$ in **Error! Reference source not found.a.**

- *The referee raised a concern about the stress amplitude effect. As the referee pointed out, the spalling strength is mainly influenced by the strain rate, and the stress amplitude plays a secondary role. For instance, a recent work suggests that the spall strength of a Zr-based MG is nearly independent of the stress amplitude [R22]. However, as the referee emphasized, whether this is universal for all MGs still needs more investigations. Thus, we have revised the relevant discussions with caution on Page 4 as follows:*

We note that the spall strength of materials may be influenced by the compressive stress amplitude which is coupled with the strain rate. Nonetheless, a recent study suggests that the spall strength of a Zr-based MG is barely dependent on the stress amplitude⁴⁰. Thus, it is reasonable to argue that the strain rate is the primary determinant influencing the material strength.

Response references

- [R1] B. Jodar, D. Loison, Y. Yokoyama, E. Lescoute, M. Nivard, L. Berthe, J.C. Sangleboeuf, Localized atomic segregation in the spalled area of a Zr₅₀Cu₄₀Al₁₀ bulk metallic glasses induced by laser-shock experiment, *Journal of Physics D: Applied Physics* 51(6) (2018).
- [R2] J.P. Escobedo, Y.M. Gupta, Dynamic tensile response of Zr-based bulk amorphous alloys: Fracture morphologies and mechanisms, *Journal of Applied Physics* 107(12) (2010).
- [R3] H. Tamura, T. Kohama, K. Kondo, M. Yoshida, Femtosecond-laser-induced spallation in aluminum, *Journal of Applied Physics* 89(6) (2001) 3520-3522.
- [R4] F. Langenhorst, M. Boustie, A. Migault, J.P. Romain, Laser shock experiments with nanoseconds pulses: a new tool for the reproduction of shock defects in olivine, *Earth and Planetary Science Letters* 173(3) (1999) 333-342.
- [R5] X.C. Tang, C. Li, H.Y. Li, X.H. Xiao, L. Lu, X.H. Yao, S.N. Luo, Cup-cone structure in spallation of bulk metallic glasses, *Acta Materialia* 178 (2019) 219-227.
- [R6] P. Guan, S. Lu, M.J. Spector, P.K. Valavala, M.L. Falk, Cavitation in amorphous solids, *Phys Rev Lett* 110(18) (2013) 185502.
- [R7] A. Ninarello, L. Berthier, D. Coslovich, Models and Algorithms for the Next Generation of Glass Transition Studies, *Physical Review X* 7(2) (2017) 021039.
- [R8] Z. Zhang, J. Ding, E. Ma, Shear transformations in metallic glasses without excessive and predefinable defects, *Proc Natl Acad Sci U S A* 119(48) (2022) e2213941119.
- [R9] V. Borovikov, M.I. Mendeleev, A.H. King, Effects of stable and unstable stacking fault energy on dislocation nucleation in nano-crystalline metals, *Modelling and Simulation in Materials Science and Engineering* 24(8) (2016) 085017.
- [R10] M.I. Mendeleev, Y. Sun, F. Zhang, C.Z. Wang, K.M. Ho, Development of a semi-empirical potential suitable for molecular dynamics simulation of vitrification in Cu-Zr alloys, *The Journal of Chemical Physics* 151(21) (2019).
- [R11] K.H. Kang, I. Sa, J.C. Lee, E. Fleury, B.J. Lee, Atomistic modeling of the Cu-Zr-Ag bulk metallic glass system, *Scripta Materialia* 61(8) (2009) 801-804.
- [R12] J.W. Wilkerson, On the micromechanics of void dynamics at extreme rates, *International Journal of*

Plasticity 95 (2017) 21-42.

[R13] J.W. Wilkerson, K.T. Ramesh, A dynamic void growth model governed by dislocation kinetics, *Journal of the Mechanics and Physics of Solids* 70 (2014) 262-280.

[R14] L. Seaman, D.R. Curran, D.A. Shockey, Computational models for ductile and brittle fracture, *Journal of Applied Physics* 47(11) (1976) 4814-4826.

[R15] T. de Ressaiguier, S. Couturier, J. David, G. Niérat, Spallation of metal targets subjected to intense laser shocks, *Journal of Applied Physics* 82(5) (1997) 2617-2623.

[R16] T.P. Remington, E.N. Hahn, S. Zhao, R. Flanagan, J.C.E. Mertens, S. Sabbaghianrad, T.G. Langdon, C.E. Wehrenberg, B.R. Maddox, D.C. Swift, B.A. Remington, N. Chawla, M.A. Meyers, Spall strength dependence on grain size and strain rate in tantalum, *Acta Materialia* 158 (2018) 313-329.

[R17] W.L. Johnson, K. Samwer, A universal criterion for plastic yielding of metallic glasses with a $(T/T_g)^{2/3}$ temperature dependence, *Phys Rev Lett* 95(19) (2005) 195501.

[R18] T. Antoun, L. Seaman, D.R. Curran, G.I. Kanel, S.V. Razorenov, A.V. Utkin, *Spall fracture*, Springer Science & Business Media 2003.

[R19] C. Li, B. Li, J.Y. Huang, H.H. Ma, M.H. Zhu, J. Zhu, S.N. Luo, Spall damage of a mild carbon steel: Effects of peak stress, strain rate and pulse duration, *Materials Science and Engineering: A* 660 (2016) 139-147.

[R20] V.I. Romanchenko, G.V. Stepanov, Dependence of the critical stresses on the loading time parameters during spall in copper, aluminum, and steel, *Journal of Applied Mechanics and Technical Physics* 21(4) (1980) 555-561.

[R21] G. Kanel, *Spall fracture: methodological aspects, mechanisms and governing factors*, *International Journal of Fracture* 163(1) (2010) 173-191.

[R22] Y. Li, X. Cheng, Z. Ma, X. Li, M. Wang, Dynamic response and damage evolution of Zr-based bulk metallic glass under shock loading, *Journal of Materials Science & Technology* 93 (2021) 119-127.

REVIEWERS' COMMENTS

Reviewer #1 (Remarks to the Author):

The authors have addressed the majority of my concerns from my previous review. Unfortunately, there was a mix-up in addressing one of my comments. I asked them to further investigate the region shown in Figure 3a instead they characterized the region shown in Figure 3b. I appreciate the effort they have put forth doing the additional characterization, but it did not answer my questions. I would like for the dark band visible in the zoomed in portion of Figure 3a to be characterized in terms of chemistry, but I will not prevent the manuscript from being accepted for this reason. With that said, I do not believe the additional characterization, of the region shown in Figure 3b, adds any value to the paper; thus, I would recommend not including it.

Reviewer #4 (Remarks to the Author):

The author has furnished essential information in response to the reviewer's inquiries; therefore, we consent to accept it.

Response to the editors and reviewers

We would like to express our gratitude to the referees for their affirmation of our work. Their suggestions have greatly improved the quality of our article. In the final revision, changes have been made in the main manuscript (text highlighted in yellow). The following are the itemized responses to the reviewers' comments.

Comments from the editors and reviewers:

Reviewer #1

The authors have addressed the majority of my concerns from my previous review. Unfortunately, there was a mix-up in addressing one of my comments. I asked them to further investigate the region shown in Figure 3a instead they characterized the region shown in Figure 3b. I appreciate the effort they have put forth doing the additional characterization, but it did not answer my questions. I would like for the dark band visible in the zoomed in portion of Figure 3a to be characterized in terms of chemistry, but I will not prevent the manuscript from being accepted for this reason. With that said, I do not believe the additional characterization, of the region shown in Figure 3b, adds any value to the paper; thus, I would recommend not including it.

- *We thank the referee for the positive feedback on our revision. Following the referee's suggestion, we have removed the EDS characterization of Fig 3b (Supplementary Fig. 8 in the last version).*

Reviewer #4

The author has furnished essential information in response to the reviewer's inquiries; therefore, we consent to accept it.

- *We appreciate the referee for approving the quality of our work.*